# An Asymptotically Optimal Approximation Algorithm for Multiobjective Submodular Maximization at Scale

**Fabian Spaeh** [1]    **Atsushi Miyauchi** [2]

## Abstract

Maximizing a single submodular set function subject to a cardinality constraint is a well-studied and central topic in combinatorial optimization. However, finding a set that maximizes multiple functions at the same time is much less understood, even though it is a formulation which naturally occurs in robust maximization or problems with fairness considerations such as fair influence maximization or fair allocation. In this work, we consider the problem of maximizing the minimum over many submodular functions, which is known as multiobjective submodular maximization. All known polynomial-time approximation algorithms either obtain a weak approximation guarantee or rely on the evaluation of the multilinear extension. The latter is expensive to evaluate and renders such algorithms impractical. We bridge this gap and introduce the first scalable and practical algorithm that obtains the best-known approximation guarantee. We furthermore introduce a novel application *fair centrality maximization* and show how it can be addressed via multiobjective submodular maximization. In our experimental evaluation, we show that our algorithm outperforms known algorithms in terms of objective value and running time.

## 1. Introduction

Due to the natural diminishing returns property, maximizing submodular functions is a central task in various fields such as optimization, machine learning, and economics (Fujishige, 2005; Nemhauser et al., 1978). In submodular maximization subject to a cardinality constraint the task is to find a set $S$ of size $|S| \leq B$ for some budget $B$ such that a submodular objective $f(S)$ is maximized. Applications range from welfare maximization to experimental design and viral marketing via influence maximization (Bilmes, 2022; Kempe et al., 2003; Krause & Guestrin, 2007). Maximizing a single monotone submodular function is well understood, and the greedy algorithm is practically efficient and obtains the best possible polynomial-time approximation guarantee of $1 - \frac{1}{e} \approx 0.63$ (Nemhauser & Wolsey, 1978). Moreover, so-called *lazy evaluations* make the greedy algorithm highly scalable (Minoux, 2005).

Another line of work studies the task where $k$ different submodular objectives have to be simultaneously maximized, which is called multiobjective submodular maximization. This is also important for many applications. For instance, in robust experimental design, the goal is to maximize a submodular function $f_\theta$, which depends on an a-priori unknown parameter $\theta$ (Krause et al., 2008). One seeks to find a set that maximizes $f_\theta$ simultaneously for all valid parameters $\theta$. Recently, many applications in artificial intelligence consider the max-min fairness objective, which often naturally falls into the multiobjective scenario. One such example is fair influence maximization, where the goal is to maximize the influence among all groups (Tsang et al., 2019).

Even though applications for the multiobjective setting are numerous and relevant, solving the problem with theoretical guarantees still presents challenges for current algorithms. Indeed, the prior work can be split into two categories of approximation algorithms: First, there are algorithms that cannot recover the $(1 - \frac{1}{e})$-approximation, but are discrete and efficient in practice. The best approximation ratio for such an algorithm is $(1 - \frac{1}{e})^2 \approx 0.40$ (Udwani, 2018). Second, there are algorithms that (almost) recover the $(1 - \frac{1}{e})$-approximation guarantee (Chekuri et al., 2010; Udwani, 2018), but solve a continuous relaxation of the problem based on the multilinear extension, which is known to be impractical (Bai et al., 2018; Chen & Kuhnle, 2024; Buchbinder & Feldman, 2024). These algorithms do not scale to larger problem instances, as the ones necessitated by the mentioned applications.

This is where our work comes in: We introduce an algorithm that (almost) achieves the $(1 - \frac{1}{e})$-approximation, but does not relax the problem to the continuous setting and

[1]Boston University, Boston, USA [2]CENTAI Institute, Turin, Italy. Correspondence to: Fabian Spaeh <fspaeh@bu.edu>, Atsushi Miyauchi <atsushi.miyauchi@centai.eu>.

*Proceedings of the 42nd International Conference on Machine Learning*, Vancouver, Canada. PMLR 267, 2025. Copyright 2025 by the author(s).

is therefore efficient in practice. This settles an open question of Udwani (2018), showing that there is a practically efficient algorithm that obtains best possible approximation ratio. Our main contributions are the following:

1. We introduce a novel algorithm for multiobjective submodular maximization. Our algorithm (almost) achieves the best possible approximation ratio of $1 - \frac{1}{e}$. Our algorithm is vastly more efficient than algorithms with comparable guarantees, which rely on a continuous relaxation and a rounding procedure. Our algorithm, instead, achieves its guarantee via a novel concentration argument.

2. Furthermore, we show how multiplicative weights updates (MWU) can further reduce the number of function evaluations, which also yields an analogue to the lazy greedy algorithm for the multiobjective problem.

3. We introduce a novel application *fair centrality maximization*, as a generalization of the well-studied task of centrality maximization with fairness consideration, and show how it can be formulated as a large-scale multiobjective submodular maximization.

4. We experimentally verify that our algorithm outperforms previous methods in terms of objective value and running time. The practical efficiency of our method allows us to be the first to solve large-scale instances while retaining quality of solutions.

## 2. Related Work

Nemhauser et al. (1978) showed that the greedy algorithm achieves a $(1-\frac{1}{e})$-approximation for cardinality-constrained monotone submodular maximization. This is tight, unless P = NP (Feige, 1998). More algorithms for this and related settings exist (Calinescu et al., 2011; Badanidiyuru & Vondrák, 2014; Mirzasoleiman et al., 2015).

Krause et al. (2008) introduced the multiobjective submodular maximization problem in the context of robust experimental design. They provided a bi-criteria algorithm and showed that the problem becomes inapproximable in the regime where $B \leq k$. Chen et al. (2017) also proved that computing a $(1-\frac{1}{e})$-approximate solution for the problem is NP-hard. When surveying approximation algorithms for the problem, we distinguish between two types of algorithms.

The first type uses a continuous relaxation of the submodular set functions $f\colon 2^V \to \mathbb{R}$ to $F\colon [0,1]^V \to \mathbb{R}$, which is called the multilinear extension. After obtaining a continuous solution via techniques from convex optimization, the solution is then rounded to output a single set. Such approaches have the benefit that they (almost) offer the best possible approximation ratio of $1 - \frac{1}{e}$. However, evaluating the multilinear extension is costly and impractical for

even moderately sized problems, as we can generally only approximate the relaxation via sampling. In their seminal work, Chekuri et al. (2010) introduced the oblivious rounding scheme *Swap Rounding* and showed how this leads to an algorithm for the multiobjective problem even subject to a matroid constraint. Later, Udwani (2018) introduced a more efficient algorithm for cardinality constraints. Tsang et al. (2019) gave an algorithm that finds a continuous solution via Frank-Wolfe, and they showed that the multilinear extension can be evaluated efficiently for influence maximization.

The second type of algorithm avoids evaluation of the multilinear extension and only operates on discrete solutions. Udwani (2018) also introduced an efficient $(1 - \frac{1}{e})^2$-approximation algorithm via MWU. The algorithm maintains a vector of weights $y \in \Delta_C$, where $C$ is the index set of multiple submodular functions and $\Delta_C$ is the probability simplex over $C$, and uses a subroutine that solves the problem of maximizing the combined submodular function $\sum_{c \in C} y_c f_c(S)$. The weights are then updated based on the individual values achieved for each objective. However, combining the solutions created in each iteration introduces an additional factor of $1 - \frac{1}{e}$ into the approximation guarantee, which is the reason that this algorithm does not retain the guarantee of continuous algorithms. Orlin et al. (2018) devised a combinatorial algorithm that almost achieves the $(1 - \frac{1}{e})$-approximation, but only for the case when $k \leq 2$.

Other efficient algorithms use the greedy paradigm, but do not obtain constant-factor approximation guarantees. The aforementioned bi-criteria algorithm by Krause et al. (2008) maximizes a modified objective function, but may violate the budget constraint in order to meet its approximation guarantee. Later Chen et al. (2017), Wilder (2018), and Anari et al. (2019) also gave bi-criteria approximation algorithms.

Instead of maximizing the minimum of multiple objectives, Soma & Yoshida (2017) considered *regret ratio minimization* where the goal is to find a collection of subsets (rather than a single subset) that approximate the Pareto optimal set to the multiple objectives. They designed algorithms for the problem which were improved by Feng & Qian (2021) and Wang et al. (2023). Malherbe & Scaman (2022) considered an objective based on quantile maximization. Recently, Wang et al. (2024) introduced the problem of maximizing the average of objectives subject to multiple constraints ensuring the value of each objective. Fazzone et al. (2024) studied a more general problem and extended the algorithm of Krause et al. (2008) to obtain a sequence of approximately Pareto optimal solutions.

Related to the multiobjective problem, submodular maximization has also been studied with fairness and diversity considerations for the output subset. In this context, each element is associated with a categorical attribute such as race or gender, and the output is required to ensure that

no particular attribute value is under- or over-represented. This problem has actively been studied in both the offline setting (Celis et al., 2018; El Halabi et al., 2024; Tang & Yuan, 2023; Yuan & Tang, 2023) and the streaming setting (El Halabi et al., 2020; 2023; Wang et al., 2021; Cui et al., 2024).

## 3. Problem Definition

We now formally define the notion of submodularity and the problem we consider in this work. A set function on a finite universe $V$ is a function $f \colon 2^V \to \mathbb{R}$ on all subsets of $V$. A set function $f$ is said to be *monotone* if $f(S) \leq f(T)$ for all sets $S \subseteq T \subseteq V$. For sets $S \subseteq V$ and $T \subseteq V$, we define the *marginal gain* of $T$ over $S$ as $f(T \mid S) = f(S \cup T) - f(S)$. If $T$ consists of a single element $v$, we simply write $f(v \mid S)$ instead of $f(\{v\} \mid S)$. A set function is called *submodular* if it has diminishing returns, i.e., for all sets $S \subseteq T \subseteq V$ and elements $v \in V$ holds that

$$f(v \mid S) \geq f(v \mid T).$$

Maximization of submodular functions is well-understood. We consider (multiplicative) approximation ratios, so we assume throughout that submodular functions are nonnegative.

In our work, we are interested in a generalization of this objective where we consider multiple submodular functions over the same universe. The goal is to find a set that maximizes all functions at the same time:

*Problem* 3.1 (Multiobjective Submodular Maximization). Let $C$ be a finite set, which we refer to as the set of colors, and define $k = |C|$. We are given a universe $V$ and monotone and submodular set functions $f_c \colon 2^V \to \mathbb{R}_{\geq 0}$ for each $c \in C$, all on the same universe $V$. Given a budget $B$, we are asked to find a set $S \subseteq V$ of size $|S| \leq B$ maximizing the least value among all colors, which is $\min_{c \in C} f_c(S)$.

In our work, we are concerned with the case when the budget is large compared to the number of colors, i.e., $B \geq k$, as this avoids the regime where the problem becomes inapproximable (Krause et al., 2008). Submodular maximization (i.e., the special case of a single color) is known to be NP-hard as it generalizes the maximum coverage problem. For the latter, no polynomial-time algorithm with an approximation ratio better than $1 - \frac{1}{e}$ is known. At the same time, the greedy algorithm provably achieves this approximation ratio for any submodular function (Nemhauser et al., 1978). Multiobjective submodular maximization (i.e., the general case of multiple colors) is a harder problem, but we aim for (almost) the same approximation guarantee. As in the prior work, our goal is to design an algorithm that provably obtains an approximation ratio of $1 - \frac{1}{e} - \epsilon$ with probability $1 - \delta$ for any pair of $\epsilon, \delta > 0$. Both values show up in the

---

**Algorithm 1:** Greedy for Problem 3.1

**Input:** Monotone and submodular set functions $f_c \colon 2^V \to \mathbb{R}_{\geq 0}$ for $c \in C$, budget $B$, and the optimal value OPT

1   $S^{(0)} \leftarrow \emptyset$;
2   **for** $i = 1, 2, \ldots, B$ **do**
3     Let $x^{(i)} \in \Delta_V$ be a solution to the problem:
4     Find $x \in \Delta_V$ such that for all $c \in C$ holds

$$\sum_{v \in V} x_v f_c(v \mid S^{(i-1)}) \geq \frac{1}{B}\left(\mathrm{OPT} - f_c(S^{(i-1)})\right);$$

5     $\qquad\qquad\qquad\qquad\qquad\qquad\qquad (1)$

6     Sample $v^{(i)} \sim x^{(i)}$;
7     Update $S^{(i)} \leftarrow S^{(i-1)} \cup \{v^{(i)}\}$;
8   **return** $S^{(B)}$.

---

running time of our algorithm and in a necessary condition on the budget $B$.

**Further Notation.** We use OPT for the optimal solution $\mathrm{OPT} = \arg\max_{S \subseteq V : |S| \leq B} \min_{c \in C} f_c(S)$ of the multiobjective submodular maximization (Problem 3.1). Abusing notation, we also use OPT to denote the optimal value, i.e., we write $\mathrm{OPT} = \min_{c \in C} f_c(\mathrm{OPT})$. Given a finite set $X$, we denote the set of probability distributions over $X$ as the simplex $\Delta_X = \{x \in [0,1]^X : \sum_{v \in X} x_v = 1\}$. We will use this to denote probability distributions over the set of colors $\Delta_C$ or the universe $\Delta_V$.

## 4. Our Algorithm

We sketch the central component of our algorithm for multiobjective submodular maximization (Problem 3.1) in Algorithm 1. Note that this is merely to illustrate and motivate our core ideas, and we also provide an implementable version as Algorithm 2. Before delving into the analysis of our approximation guarantee, we want to motivate our algorithm design. For now, we assume that we know the optimal value OPT; an assumption that we will later remove.

As in the greedy paradigm, we construct a solution $S^{(i)}$ over $1 \leq i \leq B$ iterations and in each iteration, we identify a good element $v^{(i)}$ to add. For the case of submodular maximization (i.e., the special case of a single color), using the element with maximum marginal gain $v^* = \arg\max_{v \in V} f(v \mid S^{(i-1)})$ is sufficient to obtain the $(1 - \frac{1}{e})$-approximation. However, the analysis reveals that it is not necessarily required to add the element of maximum marginal gain. Indeed, adding any element $v \in V$ whose

marginal gain satisfies the inequality

$$f(v \mid S^{(i-1)}) \geq \frac{1}{B}\left(\text{OPT} - f(S^{(i-1)})\right) \qquad (2)$$

is sufficient. Intuitively, adding an element with a marginal gain that is high enough compared to the distance to optimality is sufficient, and it is easy to show that $v^*$ also satisfies Inequality (2). Transferring this to the case of multiple colors is not straightforward as there is not necessarily a single element that satisfies Inequality (2) simultaneously for all colors $c \in C$. Instead, we show that we can satisfy (2) in expectation with a probability distribution $x^{(i)} \in \Delta_V$ which is exactly what we require for (1) in the description of Algorithm 1. As of now, it is unclear that such a probability distribution exists, but show this in the following lemma:

**Lemma 4.1.** *In each iteration of Algorithm 1, there is a solution $x^{(i)} \in \Delta_V$ that satisfies Inequality (1).*

*Proof.* Fix an iteration $1 \leq i \leq B$. Let $x^* \in [0,1]^V$ be such that $x_v^* = \frac{1}{B}$ if $v \in \text{OPT}$ and $x_v^* = 0$ otherwise. Note that $x^*$ satisfies Inequality (1) for all $c \in C$ since

$$\sum_{v \in V} x_v^* f_c(v \mid S^{(i-1)})$$

$$= \frac{1}{B} \sum_{v \in \text{OPT}} f_c(v \mid S^{(i-1)})$$

$$\geq \frac{1}{B} f_c(\text{OPT} \mid S^{(i-1)}) \qquad \text{(submodularity)}$$

$$\geq \frac{1}{B}\left(f_c(\text{OPT}) - f_c(S^{(i-1)})\right) \qquad \text{(monotonicity)}$$

$$\geq \frac{1}{B}\left(\text{OPT} - f_c(S^{(i-1)})\right). \qquad \square$$

Even though we know that a solution $x^{(i)}$ exists in each iteration, we still need to argue that we can find it efficiently. To this end, we rewrite the problem of finding $x^{(i)}$ satisfying Inequality (1) as an LP. Introducing an objective into the LP even allows us to remove the dependence on OPT:

$$\text{LP}(S^{(i-1)})\colon \ \max. \ \xi \qquad (3)$$

$$\text{s.t. } \forall c \in C \colon \sum_{v \in V} x_v \left(Bf_c(v \mid S^{(i-1)}) + f_c(S^{(i-1)})\right) \geq \xi,$$

$$\sum_{v \in V} x_v = 1,$$

$$\forall v \in V \colon x_v \geq 0.$$

Note that the equivalence with (1) follows immediately by rearranging terms in the first constraint using the second constraint $\sum_{v \in V} x_v = 1$ and replacing $\xi$ with OPT. It is thus guaranteed by Lemma 4.1 that the optimal value $\xi^*$ of the LP satisfies $\xi^* \geq \text{OPT}$.

Now, operating over continuous solutions $x^{(i)}$ is impractical as it involves relaxing the submodular functions. On

---

**Algorithm 2:** LP Greedy with Independent Repetitions

**Input:** Monotone and submodular set functions $f_c \colon 2^V \to \mathbb{R}_{\geq 0}$ for $c \in C$, budget $B$, failure probability $\delta$

1   $S_{\text{best}} \leftarrow \emptyset$;

2   **for** $t = 1, 2, \ldots, \lceil \log(2/\delta) \rceil$ **do**

3     $S^{(0)} \leftarrow \emptyset$;

4     **for** $i = 1, 2, \ldots, B$ **do**

5       Let $x^{(i)} \in \Delta_V$ be a solution to $\text{LP}(S^{(i-1)})$ ;

6       Sample $v^{(i)} \sim x^{(i)}$;

7       Update $S^{(i)} \leftarrow S^{(i-1)} \cup \{v^{(i)}\}$ ;

8     **if** $\min_{c \in C} f_c(S^{(B)}) \geq \min_{c \in C} f_c(S_{\text{best}})$ **then**

9       $S_{\text{best}} \leftarrow S^{(B)}$ ;

10   **return** $S_{\text{best}}$.

---

the contrary, we are able to avoid continuous solutions as we discretize immediately by sampling a random element $v^{(i)} \in V$ according to $x^{(i)}$ and adding it to our solution. The technical part of our analysis is concerned with showing that this leads to a solution of high value for all colors $c \in C$, with sufficiently high probability. However, intuitively this is clear: Assume we have constructed a partial solution $S^{(i)}$ that has low value for a specific color $c \in C$. This means that the gap to optimality $\text{OPT} - f_c(S^{(i-1)})$ is larger for $c$ than for other colors, so the solution $x^{(i-1)} \in \Delta_V$ will put more mass on elements $v \in V$ that satisfy (1) for the color $c$. This makes it more likely that $f_c(S^{(i)})$ will be larger, and overall that all colors have high value, given that the budget is sufficiently large.

Finally, we need to amplify the success probability of our algorithm, which we do via independent repetitions of Algorithm 1. We detail our complete approach in Algorithm 2, which also no longer requires knowledge of OPT.

Adding random elements has been considered in prior works on submodular maximization such as in the algorithm of Buchbinder et al. (2014) for non-monotone submodular maximization, where the randomness helps to avoiding bad elements. However, our algorithm and sampling is carefully designed to solve the multiobjective problem: We choose our sampling probabilities to enforce progress across all colors simultaneously, which can be understood as a discrete version of the continuous approach of Chekuri et al. (2010).

### 4.1. Analysis Outline

We now state the approximation guarantee of Algorithm 1. This guarantee only holds if the optimal value OPT is sufficiently large, but we will show in Section 4.2 how to convert this into a guarantee that depends on the ratio of the budget $B$ and the number of colors $k = |C|$ via a pre-processing

step. To bound the running time and the number of function evaluations of our algorithm, we will show in Section 4.3 how to use MWU to solve the LP efficiently. We defer some of our proofs to Appendix A.

**Theorem 4.2.** *If* $\text{OPT} \geq \frac{4}{\epsilon^2} M \log(2k)$ *where the maximum marginal gain is* $M = \max_{c \in C} \max_{v \in V} f_c(v \mid \emptyset)$, *then Algorithm 2 outputs a solution $S$ such that*

$$\min_{c \in C} f_c(S) \geq \left(1 - \frac{1}{e} - \epsilon\right) \text{OPT}$$

*with probability at least $1 - \delta$.*

To prove the theorem, we begin with the basic observation that in each iteration, we add an element to the solution that for each $c \in C$ is good in expectation.

**Lemma 4.3.** *In each iteration $i$ of Algorithm 1 and for each color $c \in C$,*

$$\mathbb{E}[f_c(v^{(i)} \mid S^{(i-1)}) \mid S^{(i-1)}] \geq \frac{1}{B}\left(\text{OPT} - f_c(S^{(i-1)})\right).$$

In expectation, we recover the greedy guarantee:

**Lemma 4.4.** *Algorithm 1 outputs a solution $S$ such that for each $c \in C$,*

$$\mathbb{E}[f_c(S)] \geq \left(1 - \frac{1}{e}\right) \text{OPT}.$$

The problem is that this is only in expectation, but we have to satisfy it for all colors $c \in C$ simultaneously. To this end, we show how to use concentration results for martingales.

To motivate our analysis further we explain how this generalizes an easier problem that is better understood: Imagine that in each iteration $i$, we obtain $x^{(i)} = x^*$ as defined in the proof of Lemma 4.1 as a solution to (1). Since this solution does not change over the iterations, our algorithm is equivalent to throwing $B$ balls (one ball per iteration) into $B$ bins (one bin per element of OPT). It is known that the resulting number of non-empty bins is $1 - \frac{1}{e}$ in expectation. In fact, we have shown a stronger statement in Lemma 4.4, i.e., that this even holds for functions that are submodular over the non-empty bins. Our analysis thus takes inspiration from concentration results for this balls-into-bins problem to obtain concentration. In particular, we use a form of Azuma's inequality due to Kuszmaul & Qi (2021):

**Theorem 4.5** (Corollary 16 in Kuszmaul & Qi (2021)). *Suppose that Alice constructs a sequence of random variables $X_1, \dots, X_B$, with $X_i \in [0, M]$, $M > 0$, using the following iterative process. Once the outcomes of $X_1, \dots, X_{i-1}$ are determined, Alice then selects the probability distribution $\mathcal{D}_i$ from which $X_i$ will be drawn; $X_i$ is then drawn from distribution $\mathcal{D}_i$. Alice is an adaptive adversary in that she can adapt $\mathcal{D}_i$ to the outcomes of $X_1, \dots, X_{i-1}$. The*

*only constraint on Alice is that $\sum_{i=1}^{B} \mathbb{E}[X_i \mid \mathcal{D}_i] \geq \mu$, that is, the sum of the means of the probability distributions $\mathcal{D}_1, \dots, \mathcal{D}_B$ must be at least $\mu$. If $X = \sum_{i=1}^{B} X_i$, then for any $\epsilon > 0$,*

$$\Pr[X \leq (1 - \epsilon)\mu] \leq \exp\left(-\frac{\epsilon^2 \mu}{2M}\right).$$

We use this to argue for concentration around (the lower bound on) the expectation, which was $(1 - \frac{1}{e})\text{OPT}$:

**Lemma 4.6.** *Let $S$ be the output of Algorithm 1. For each $c \in C$, it holds that*

$$\Pr\left[f_c(S) \leq \left(1 - \frac{1}{e} - \epsilon\right)\text{OPT}\right] \leq \exp\left(-\frac{\epsilon^2 \text{OPT}}{4M_c}\right)$$

*where $M_c = \max_{v \in V} f_c(v \mid \emptyset)$.*

*Proof.* Fix $c \in C$. In the context of Theorem 4.5, we use $X_i = f_c(v^{(i)} \mid S^{(i-1)})$ and the distribution $\mathcal{D}_i$ over values $f_c(v^{(i)} \mid S^{(i-1)})$ where $v^{(i)} \sim x^{(i)}$. As such, $M = \max_{c \in C} M_c$ is also the maximum marginal value. Note that the random process is exactly as in Algorithm 1: In each iteration, we select a (possibly adversarial) solution $x^{(i)}$ to (1) which determines the distribution $\mathcal{D}_i$. It remains to argue that $\sum_{i=1}^{B} \mathbb{E}[X_i \mid \mathcal{D}_i] \geq \mu$ for $\mu = (1 - \frac{1}{e})\text{OPT}$. However, we have just shown this in Lemma 4.4:

$$\sum_{i=1}^{B} \mathbb{E}[X_i \mid \mathcal{D}_i] = \mathbb{E}\left[\sum_{i=1}^{B} f_c(v^{(i)} \mid S^{(i-1)})\right]$$
$$= \mathbb{E}[f_c(S^{(B)})] \geq \left(1 - \frac{1}{e}\right)\text{OPT} = \mu.$$

Applying Theorem 4.5 now yields

$$\Pr[f_c(S) \leq (1 - \epsilon)\mu] \leq \exp\left(-\frac{\epsilon^2 \mu}{2M_c}\right).$$

Noticing that $\left(1 - \frac{1}{e} - \epsilon\right)\text{OPT} \leq (1 - \epsilon)\mu$ and $1 - \frac{1}{e} > \frac{1}{2}$, we have the lemma. $\square$

With the concentration for each color in hand, we are now ready to prove Theorem 4.2:

*Proof of Theorem 4.2.* Recall that we defined the maximum gain as $M = \max_{c \in C} M_c = \max_{c \in C} \max_{v \in V} f_c(v \mid \emptyset)$. Assuming $\text{OPT} \geq \frac{4}{\epsilon^2} M \log(2k)$, we have by Lemma 4.6 for each $c \in C$ that

$$\Pr\left[f_c(S) \leq \left(1 - \frac{1}{e} - \epsilon\right)\text{OPT}\right]$$
$$\leq \exp\left(-\frac{\epsilon^2 \text{OPT}}{4M}\right) \leq \frac{1}{2k}$$

and by a union bound over the $k$ (dependent) colors, we obtain that the result holds for all $c \in C$ with probability $\geq \frac{1}{2}$. We repeat the algorithm $r = \lceil \log(1/\delta) \rceil$ times and output the best solution. Since repetitions are independent, the probability that none of the $\lceil \log(1/\delta) \rceil$ solutions is a $(1 - \frac{1}{e} - \epsilon)$-approximation is $2^{-r} \leq e^{-\log(1/\delta)} = \delta$. $\square$

### 4.2. Removing the Condition on OPT

We can push the condition on OPT into a condition on the relation of the budget $B$ to the number of colors $k = |C|$, which we state in Theorem 4.8. As do all $(1 - \frac{1}{e} - \epsilon)$-approximation algorithms for Problem 3.1, we also require the budget to be large. Recall that this avoids the regime where the problem becomes inapproximable (Krause et al., 2008) and is natural for many of the applications we mentioned in the introduction. Our condition is slightly less restrictive compared to the prior work, and we provide a detailed comparison at the end of this section.

The key is a pre-processing step where we eliminate elements $v \in V$ whose value $f_c(v \mid \emptyset)$ is large compared to OPT for any color $c \in C$, as such elements impair the concentration. We find such elements and add them to a partial solution $T$. Afterwards, some colors may have reached OPT. However, we now only consider the colors

$$\tilde{C} = \{c \in C : f_c(T) < \text{OPT}\}$$

which have not yet reached OPT. Our pre-processing guarantees that marginal gains for colors $c \in \tilde{C}$ on top of the partial solution $T$ are small compared to OPT, and are thus not problematic for Algorithm 2. This is similar to the pre-processing step described by Udwani (2018), but our pre-processing does not require knowledge of OPT. We defer the full description of the pre-processing routine to Algorithm 3 in Appendix A.2, along with all proofs.

We then run Algorithm 2 with a reduced budget $B - |T|$ on functions $\tilde{f}_c(A) = f_c(A \cup T)$, which ensures by submodularity that marginal gains are small for the modified instance. The technical difficulty is to show that reducing the budget does not result in a big decrease of the optimum, which is non-trivial for multiple objectives. We provide the following novel result:

**Lemma 4.7.** *Let* $\widetilde{\text{OPT}}_b = \max_{S \subseteq V : |S| \leq b} \min_{c \in \tilde{C}} \tilde{f}_c(S)$ *for* $b \geq 0$. *Let* $\gamma > 0$ *be such that* $\tilde{f}_c(v \mid \emptyset) \leq \gamma \widetilde{\text{OPT}}_B$ *for all* $v \in V$ *and* $c \in \tilde{C}$. *Then, for* $\tilde{B} \leq B$,

$$\widetilde{\text{OPT}}_{\tilde{B}} \geq \left(1 - \sqrt{3\gamma \frac{B}{\tilde{B}} \log k}\right) \frac{\tilde{B}}{B} \widetilde{\text{OPT}}_B.$$

Note that prior work uses similar results but over the continuous space, which makes our result more challenging. Let $\tilde{S}$ be the solution of Algorithm 2 to the modified instance. We have the following guarantee on the combined solution:

**Theorem 4.8.** *If* $B \geq 108 \frac{k}{\epsilon^3} \log k$ *then*

$$\min_{c \in C} f_c(T \cup \tilde{S}) \geq \left(1 - \frac{1}{e} - \epsilon\right) \text{OPT}$$

*with probability at least* $1 - \delta$.

**Comparison with prior work.** We compare our condition $B = \Omega(\frac{k}{\epsilon^3} \log k)$ of Theorem 4.8 with conditions on the budget of prior work. Udwani (2018) requires that $\epsilon \leq \frac{1}{10 \log k}$ for an approximation ratio of $1 - \frac{1}{e} - \epsilon - \frac{k}{B\epsilon^3} - (\frac{\log B}{B})^{1/2}$. In order to achieve an approximation ratio of $1 - \frac{1}{e} - \epsilon$, his algorithm requires that $B = \Omega(\frac{k}{\epsilon^4})$. The algorithm of Tsang et al. (2019) provides a similar approximation that is also dominated by the term $\frac{k}{B\epsilon^3}$. Since they have to use the same choice of $\epsilon$, their algorithm has the same condition on $B$. The condition on $\epsilon$ in both works requires $\frac{1}{\epsilon} \geq 10 \log k$, which means that our condition for Theorem 4.8 is never more restrictive, while our condition is strictly less restrictive in its dependence on $\epsilon$.

### 4.3. MWU and Lazy Evaluations

Algorithm 2 may still be impractical. First, solving the LP may be expensive and second, we need to evaluate all marginal gains in each iteration. The latter is also a problem appearing in the greedy algorithm for a single color and we there resort to lazy evaluations. It turns out that we can solve both efficiency problems via multiplicative weights updates (MWU) (Arora et al., 2012). In particular, we show that solving the LP approximately via a few rounds of MWU is sufficient, and we even transfer lazy evaluations to multiple colors. We note that MWU have been used in the context of multiobjective submodular maximization, but to solve a continuous relaxation of the problem (Chekuri et al., 2015).

We detail this in Algorithm 4 in Appendix A.3. We crucially use that solving LP($S$) admits a formulation as a two-player game: The first player selects a probability distribution $y \in \Delta_C$ over the colors, and the second player responds with the element $v \in V$ which maximizes the convex combination $\sum_{c \in C} y_c \ell_c(v)$ where $\ell_c(v)$ is a loss (or rather, a gain) given as $\ell_c(v) = B f_c(v \mid S) + f_c(S)$. Finding the best response thus entails a search over all elements $v \in V$, and we show how to transfer the concept of lazy evaluations to the multiobjective problem in order to reduce the number of function evaluations in practice. Furthermore, two-player games can be solved efficiently using few rounds of MWUs (Arora et al., 2012). The complication is to bound the number of rounds, and we do so by appealing to the structure of our problem. This allows us to bound the overall running time of our algorithm. Note that we first run the pre-processing described in Algorithm 3, and then apply Algorithm 2 where we solve the LP approximately using MWU as described in Algorithm 4.

**Theorem 4.9.** *Our algorithm for multiobjective submodular maximization runs in time $O(nB^3 \frac{1}{\epsilon^2} k \log(k) \log(1/\delta))$ and requires $O(nBk \log(1/\delta))$ function evaluations.*

We also defer this proof to Appendix A.3. We compare this with the $(1 - \frac{1}{e} - \epsilon)$-approximation algorithm of Udwani (2018), which has a similar condition on $B$ but needs $\tilde{O}(kn^8)$ function evaluations. The faster $((1 - \frac{1}{e})^2 - \epsilon)$-approximation algorithm requires $\tilde{O}(n\frac{1}{\epsilon^3})$ function evaluations by repeatedly solving submodular maximization problems using a nearly-linear time algorithm developed by Mirzasoleiman et al. (2015). Furthermore, the $(1 - \frac{1}{e} - \epsilon)$-approximation algorithm of Tsang et al. (2019) needs $\tilde{O}(\frac{kB^2}{\epsilon} + \frac{B^4}{\epsilon^5})$ evaluations of the multilinear extension and its gradient, and needs $O(\frac{nB^2}{\epsilon^2} + \frac{kB^2}{\epsilon} + \frac{B^3}{\epsilon^2})$ additional time. Note that the evaluation of the multilinear extension is generally not tractable and usually estimated via sampling (Salem et al., 2024). Only special structure of the submodular functions allows for more efficient evaluations.

## 5. Fair Centrality Maximization

We now introduce our novel application which furthers the applicability of algorithms for multiobjective submodular maximization. Centrality measures, quantifying the importance of nodes or edges in a network, play a key role in network analysis (Das et al., 2018). In many real-world scenarios, we want to optimize the centrality score of a target node by intervening in the structure of the network (e.g., by adding or removing edges around the node). In particular, *centrality maximization* is a well-studied task, where given a directed graph $G = (V, A)$, a target node $v \in V$, and a budget $B \in \mathbb{Z}_{>0}$, we are asked to insert at most $B$ edges heading to $v$ that maximize the centrality score of $v$ (see e.g., (Bergamini et al., 2018; Crescenzi et al., 2016; D'Angelo et al., 2019; Ishakian et al., 2012; Medya et al., 2018)).

Among existing centrality measures, the *harmonic centrality* is one of the most well-established (Boldi & Vigna, 2014). The harmonic centrality score of $v \in V$ is defined as

$$h_G(v) = \sum_{u \in V \setminus \{v\}} \frac{1}{d_G(u, v)}$$

where $d_G(u, v)$ is the shortest-path distance from $u$ to $v$ on $G$. Intuitively, the score quantifies the importance of nodes based on the level of reachability from the other nodes. Boldi & Vigna (2014) showed that unlike previously known centrality measures, the harmonic centrality satisfies all the desirable axioms, namely the size axiom, density axiom, and score monotonicity axiom. Recently, Murai & Yoshida (2019) theoretically and empirically demonstrated that among well-known centrality measures, the harmonic centrality is most stable and thus reliable against the structural uncertainty of networks.

In many real-world networks, nodes are not uniform but varied in terms of attributes. Examples include social networks, where nodes have sensitive attributes such as race, gender, religion, or even political opinions. Suppose that we have a network in which each node has a categorical attribute such as race or gender. Centrality maximization without any consideration of the variation of node attributes may lead to undesirable outcomes. Indeed, as the objective function, i.e., the centrality measure employed, does not take into account the variation of node attributes, it cannot distinguish between parts of centrality scores of the target node corresponding to nodes with different attribute values. Therefore, even if the centrality score of the target node is maximized, the node might still not be sufficiently visible to nodes with some specific attribute value. This is problematic, for instance, in the case where we use centrality maximization to improve the visibility of public health agency's accounts in social media platform. Such accounts should be sufficiently visible even to minority users.

We therefore study centrality maximization with fairness considerations, taking into account the variation of node attributes. To this end, we introduce a novel centrality measure, which we refer to as the *fair harmonic centrality*. This measure is a generalization of the aforementioned harmonic centrality, which contributes to finding a fair solution in terms of node attributes. Let now $C$ be a set of colors, i.e., attribute values. Let $\ell \colon V \to C$ be a mapping that assigns each node to a color. For each $c \in C$, define $V_c = \{v \in V \mid \ell(v) = c\}$. We define the fair harmonic centrality based on the maximin fairness (Rahmattalabi et al., 2019) as follows:

$$h_G^{\min}(v) = \min_{c \in C} \frac{1}{|V_c \setminus \{v\}|} \sum_{u \in V_c \setminus \{v\}} \frac{1}{d_G(u, v)}. \quad (4)$$

This represents the minimum value among all parts of the harmonic centrality score of $v$ corresponding to nodes with different colors, normalized by their populations. Clearly, the above is a generalization of the original harmonic centrality. Based on this measure, we formulate:

*Problem* 5.1 (Fair Centrality Maximization). Given a directed graph $G = (V, A)$, a mapping $\ell \colon V \to C$, a target node $v \in V$, and a budget $B \in \mathbb{Z}_{>0}$, we are asked to find $F \subseteq \{u \in V \mid (u, v) \notin A\} \times \{v\}$ with $|F| \leq B$ that maximizes $h_{(V, A \cup F)}^{\min}(v)$.

This is a special case of the multiobjective submodular maximization (Problem 3.1), since every term in the minimum in (4) is submodular in $F$, as in the original harmonic centrality (Crescenzi et al., 2016). Existing algorithms for multiobjective submodular maximization either offer weak theoretical guarantees or rely on the multilinear extension of the objective function; however, the evaluating the latter is computationally expensive when considering our fair centrality

maximization problem for large-scale networks. With our algorithm, we are the first to solve multiobjective submodular maximization on this scale with the strongest guarantee.

# 6. Experimental Evaluation

We evaluate our algorithm, natural greedy baselines, and other prior work on synthetic and real-world instances for max-$k$-cover, fair centrality maximization, and fair influence maximization. We run our experiments in Python 3 on a ThinkPad X1 Carbon with an Intel Core i7-1165G7 CPU and 16GB of RAM. Our code is publicly available.[1]

## 6.1. Algorithms

We use our algorithm and several baseline algorithms with and without theoretical guarantees. For baselines that require an estimate $\mathrm{OPT}'$ to the optimal value, we use an outer loop that performs a binary search for $\mathrm{OPT}'$. Note that our algorithm does not require such an estimate.

**LP Greedy.** We use our Algorithm 2 with 20 repetitions of the outer loop to boost the success probability, and solve LP (3) using Gurobi Optimizer 11.0.1 (Gurobi Optimization, LLC, 2024). We facilitate lazy evaluations differently from the MWU described in Algorithm 4: In each iteration $i$, we solve LP($S^{(i-1)}$), but using the upper bounds $g_c(v)$ in place of the real marginal gains. Whenever the solution $x \in \Delta_V$ to the LP places mass $x_v > 0$ on an element $v \in V$, we evaluate and update $g_c(v) = f_c(v \mid S^{(i-1)})$. We then solve the LP again and repeat this until the real marginal gains of all elements $v \in V$ with $x_v > 0$ are evaluated. We omit the pre-processing described in Algorithm 3. Algorithm 2 may act overly conservative in its selection to preserve the approximation ratio for difficult instances. We thus modify the left-hand-side of the first constraint of LP (3) to be

$$\sum_{v \in V} x_v \left( B f_c(v \mid S^{(i-1)}) + \phi f_c(S^{(i-1)}) \right)$$

where $\phi > 1$ is a factor that controls the greediness of our algorithm: For larger $\phi$, our algorithm prefers to increase the color that currently has the least value, instead of picking a distribution $x$ that leads to a balanced increase. Throughout, we use a factor $\phi = 10$. We include an ablation study for the number of repetitions of the outer loop and the factor $\phi$.

**Greedy heuristics.** We use two greedy heuristics. First, we use the heuristic described by Udwani (2018) (GREEDY ROUND ROBIN): In $1 \le i \le B$ iterations, we add one element after another to the solution $S^{(i-1)}$. We select the $i$-th element as $v^{(i)} = \arg\max_{v \in V} f_{c^*}(v \mid S^{(i-1)})$ where $c^* = i \mod k$. We use another greedy heuristic (GREEDY MINIMUM), where $c^* = \arg\min_{c \in C} f_c(S^{(i-1)})$.

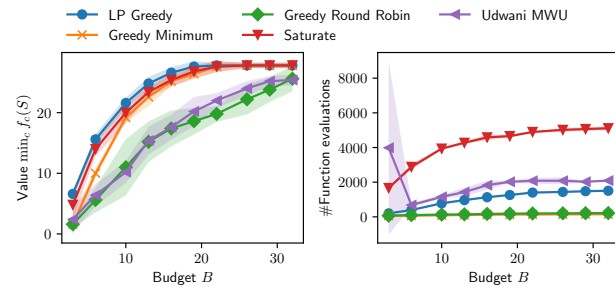

*Figure 1.* Multiobjective submodular maximization for max-$k$-cover. We use $k = 20$ Kronecker graphs on $n = 64$ nodes. We show the function value (left) and the number of evaluations (right). We report mean and standard deviation over 5 random instances.

**Saturate.** We use the bi-criteria algorithm SATURATE due to Krause et al. (2008) as a heuristic with fixed budget. The algorithm uses a guess $\mathrm{OPT}'$ to the optimal value and optimizes greedily over the submodular function $f_{\text{SATURATE}}(S) = \sum_{c \in C} \min\{f_c(S), \mathrm{OPT}'\}$.

**Udwani's MWU.** We use the efficient $(1 - \frac{1}{e})^2$-approximation described by Udwani (2018) (UDWANI MWU). We follow his implementation details and also omit the pre-processing. We use 100 iterations with a step size of $\eta = 0.1 \cdot \mathrm{OPT}'$. We use the lazy greedy implementation to solve the inner maximization problem.

**Continuous Frank-Wolfe for influence maximization.** We use the algorithm of Tsang et al. (2019) and their implementation. The algorithm relies on the fast evaluation of the multilinear extension for fair influence maximization, and thus we use it only for fair influence maximization.

## 6.2. Max-$k$-Cover

First, we replicate the setup of Udwani (2018) for max-$k$-cover instances: Here, we generate $k$ random synthetic graphs $\{G_c = (V, E_c)\}_{c \in C}$ on a fixed vertex set $V$. The cover size of $U \subseteq V$ on graph $G_c$ is $f_c(U) = |N_{G_c}(U)|$ where $N_{G_c}(U) = \{\{u, v\} \in E_c : u \in U \text{ or } v \in U\}$. We use $k = 20$ stochastic Kronecker graphs on $n = 64$ nodes which reflect real-world networks and are detailed in Udwani (2018) and Leskovec et al. (2010). Our results are in Figure 1 and we include further experiments on other graphs in Appendix B.1. This shows that we achieve the highest objective with fewer function evaluations than other algorithms, particularly compared with UDWANI MWU which is the only other algorithm with theoretical guarantees.

## 6.3. Fair Centrality Maximization

We use Amazon co-purchasing networks used by Anagnostopoulos et al. (2020) and Miyauchi et al. (2023) along with the color attributes which represent product categories to ob-

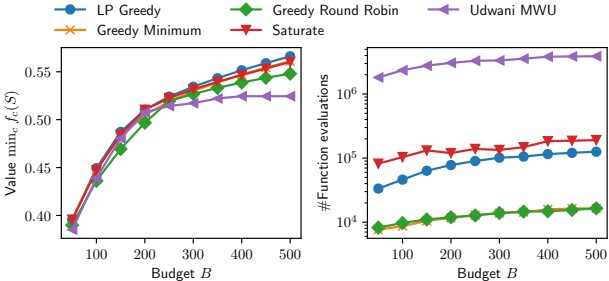

**Figure 2.** Fair centrality maximization on the Amazon co-purchasing graph *Arts, Crafts & Sewing* with $n = 5051$ nodes and $k = 2$ colors.

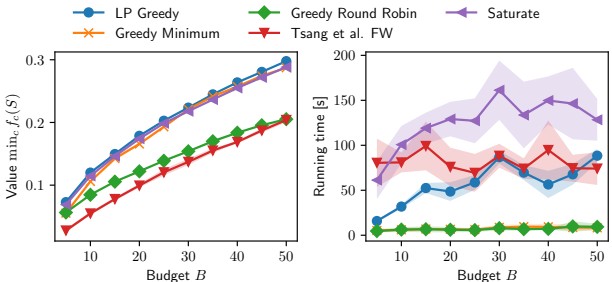

**Figure 3.** Fair influence maximization on a simulated Antelope Valley network of $n = 500$ nodes on attribute *ethnicity* with $k = 5$.

tain instances for our fair centrality maximization problem as defined in Section 5. The networks are available online (tsourakakis-lab, 2024) and contain large graphs of up to 10 380 nodes and 53 680 edges. We select an arbitrary target node among the nodes with median degree. Figure 2 shows results for a single network called *Arts, Crafts & Sewing* and we include results for other networks in Appendix B.2. Throughout, our algorithm achieves the highest objective value with fewer function evaluations than UDWANI MWU and SATURATE, which are the strongest competitors. This is mainly because, compared to these two algorithms, our method does not require a binary search to determine a good guess to the optimal value.

### 6.4. Fair Influence Maximization

Finally, we replicate the setup of Tsang et al. (2019) for their influence maximization objective, where the diffusion follows the independent cascade model (Kempe et al., 2003). They define an instance for multiobjective submodular maximization via the colored influence $f_c(S) = \frac{1}{|V_c|} \mathbb{E}[\text{number of nodes with color } c \text{ that } S \text{ activates}]$. We use their simulated Antelope Valley networks on $n = 500$ nodes which can be colored according to different node attributes. As in Tsang et al. (2019), we use 1 000 samples to approximate the influence. Figure 3 shows results for the attribute *ethnicity*, where we outperform the prior work

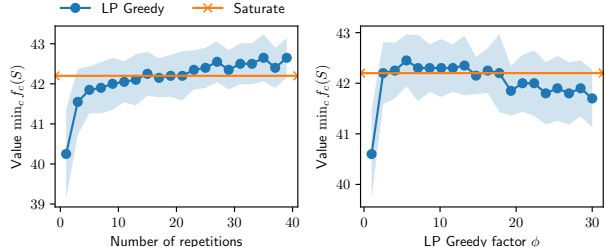

**Figure 4.** We run Algorithm 2 on a max-$k$-cover instance of $k = 20$ Erdős-Rényi random graphs on $n = 64$ nodes with $p = 0.1$. For the left plot, we vary the number of repetitions while using $\phi = 10$. For the right plot, we vary the factor $\phi$ while using 20 repetitions. We report mean and standard deviation over 5 runs.

especially for large budgets. We omit UDWANI MWU as it exceeds 10 minutes per run. The algorithm of Tsang et al. (2019) uses special evaluation oracles for the gradient of the influence which do not correspond to function evaluations, so we report the running time instead. Recall that their algorithm is specialized to influence maximization, for which we can evaluate the multilinear extension efficiently. Our algorithm applies to any multiobjective instance. Appendix B.3 contains results for further attributes, which includes one instance where the heuristic SATURATE outperforms our algorithm for a small budget. Results for the remaining graphs in Tsang et al. (2019) are similar, so we omit them.

### 6.5. Ablation Study

Figure 4 provides an ablation study for the number of repetitions and the factor $\phi$ which makes the algorithm act less conservatively. We use a simple synthetic instance of Erdős-Rényi random graphs for the max-$k$-cover problem. Our results show that after 20 repetitions, our algorithm has surpassed the objective value of SATURATE and more repetitions do only contribute to a slight increase in objective value. The right plot shows that a factor of approximately $\phi \in [5, 15]$ results in the highest objective value.

## 7. Conclusion

We introduce the first scalable algorithm for multiobjective submodular maximization that achieves a $(1 - \frac{1}{e} - \epsilon)$-approximation ratio. Going beyond fair centrality maximization and influence maximization, many problems in fairness naturally admit formulations as multiobjective problems. The high scalability and theoretical guarantees of our algorithm make it a well-suited option to address such problems. Further, we avoided the common methodology of optimizing in the continuous space and rounding the resulting solution. Our novel techniques may lead to improvements in areas even beyond the multiobjective problem.

## Impact Statement

This paper presents work whose goal is to advance the field of Machine Learning. There are many potential societal consequences of our work, none of which we feel must be specifically highlighted here.

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

# A. Omitted Details and Proofs

## A.1. Missing Proofs from Section 4.1

**Lemma 4.3.** *In each iteration $i$ of Algorithm 1 and for each color $c \in C$,*

$$\mathbb{E}[f_c(v^{(i)} \mid S^{(i-1)}) \mid S^{(i-1)}] \geq \frac{1}{B}\left(\text{OPT} - f_c(S^{(i-1)})\right).$$

*Proof.* This follows directly by our algorithm design and the definition of expectation:

$$
\begin{aligned}
\mathbb{E}[f_c(v^{(i)} \mid S^{(i-1)}) \mid S^{(i-1)}] &= \sum_{v \in V} \Pr[v^{(i)} = v \mid S^{(i-1)}] f_c(v \mid S^{(i-1)}) \\
&= \sum_{v \in V} x_v^{(i)} f_c(v \mid S^{(i-1)}) \\
&\geq \frac{1}{B}\left(\text{OPT} - f_c(S^{(i-1)})\right).
\end{aligned}
$$

$\square$

**Lemma 4.4.** *Algorithm 1 outputs a solution $S$ such that for each $c \in C$,*

$$\mathbb{E}[f_c(S)] \geq \left(1 - \frac{1}{e}\right)\text{OPT}.$$

*Proof.* We do the unrolling as in the standard greedy analysis, but in expectation over the result of Lemma 4.3. That is, Lemma 4.3 is equivalent to

$$
\begin{aligned}
\mathbb{E}[f_c(S^{(i)}) - \text{OPT} + \text{OPT} - f_c(S^{(i-1)}) \mid S^{(i-1)}] &\geq \frac{1}{B}\left(\text{OPT} - f_c(S^{(i-1)})\right) \\
\iff \mathbb{E}[\text{OPT} - f_c(S^{(i)}) \mid S^{(i-1)}] &\leq \left(1 - \frac{1}{B}\right)\left(\text{OPT} - f_c(S^{(i-1)})\right).
\end{aligned}
$$

Now, unrolling this yields

$$
\begin{aligned}
\mathbb{E}[\text{OPT} - f_c(S^{(B)})] &= \mathbb{E}_{S^{(B-1)}}\left[\mathbb{E}_{v^{(B)}}[\text{OPT} - f_c(S^{(B)}) \mid S^{(B-1)}]\right] \\
&\leq \left(1 - \frac{1}{B}\right)\mathbb{E}_{S^{(B-1)}}\left[\text{OPT} - f_c(S^{(B-1)})\right] \\
&\leq \cdots \\
&\leq \left(1 - \frac{1}{B}\right)^B \text{OPT} \\
&\leq \frac{1}{e}\text{OPT},
\end{aligned}
$$

which is equivalent to $\mathbb{E}[f_c(S^{(B)})] \geq (1 - \frac{1}{e})\text{OPT}$. $\square$

## A.2. Details about Removing the Condition on $\text{OPT}$

Before going into the analysis of Algorithm 3 we want to again motivate our algorithm design. Instances can violate the condition $\text{OPT} \geq \frac{4}{\epsilon^2} M \log(2k)$ only in two ways. First, if the budget is small, it may happen that the optimum solution is not much larger than $M$. For instance, in the extreme case in which $B = 1$, we also have $\text{OPT} = M$ and necessarily violate the condition for OPT. Second, even if we have a large budget, the optimum solution value may still not be much larger than $M$ due to submodularity, even for a single color. However, we will show that the latter case is not an issue, and we only require a sufficiently large budget. This rests on the fact that cases in which $M$ is large compared to OPT, there are only few elements with large marginal gain. We can find those elements in a pre-processing step and add them to our solution

---

**Algorithm 3:** Pre-Processing for Algorithm 2

---

**Input:** Monotone and submodular set functions $f_c \colon 2^V \to \mathbb{R}$ for $c \in C$ and per-color budget $B'$

1   $T \leftarrow \emptyset$;
2   **for** $c \in C$ **do**
3      **for** $i = 1, 2, \ldots, B'$ **do**
4          Let $v = \arg\max_{v \in V} f_c(v \mid T)$ ;
5          Update $T \leftarrow T \cup \{v\}$ ;

6   **return** $T$.

---

before running Algorithm 2. The remaining elements can only have low marginal gain, which then allows us to obtain a better concentration.

Recall that $\tilde{C} = \{c \in C : f_c(T) < \text{OPT}\}$ where $T$ is the output of Algorithm 3. Recall also that we define a modified instance where $\tilde{f}_c(A) = f_c(A \cup T)$ for all $c \in \tilde{C}$ and $A \subseteq V$.

**Lemma A.1.** *Algorithm 3 outputs a set $T$ of size $|T| \leq kB'$. Furthermore, $\tilde{f}_c(v \mid \emptyset) \leq \frac{\text{OPT}}{B'}$ for all $c \in \tilde{C}$ and $v \in V$.*

*Proof.* By the definition of Algorithm 3, we add at most $B'$ elements per color, meaning the set $T$ has size at most $kB'$. To show the second claim, let now $c \in \tilde{C}$ and $v \in V$ be any element. If $c \in T$ then $\tilde{f}_c(v \mid \emptyset) = f_c(v \mid T) = 0$ and we are done. We may thus assume that we did not add $c$ to $T$. For the proof, let us now denote with $T_c$ the elements that are added to $T$ for color $c$, and let $T_c^{(i)}$ denote $T_c$ at the end of iteration $i$. Since we did not add $v$ to $T_c$, the greedy selection implies that $f_c(v \mid T) \leq f_c(v \mid T^{(i-1)}) \leq f_c(v^{(i)} \mid T^{(i-1)})$ for all $1 \leq i \leq B'$. Hence,

$$f_c(v \mid T) \leq \frac{1}{B'} \sum_{i=1}^{B'} f_c(v^{(i)} \mid T^{(i-1)}) = \frac{1}{B'} f_c(T^{(B')} \mid T^{(0)}) \leq \frac{1}{B'} f_c(T) \leq \frac{1}{B'} \text{OPT}$$

where the penultimate inequality is by submodularity, and the last inequality due to $c \in \tilde{C}$.     $\square$

In the end, we want to show that $T \cup \tilde{S}$ attains a good fraction of the optimum solution. Clearly, Lemma A.1 shows that we can effectively reduce the maximum singleton marginal gain which is sufficient to improve the concentration in Theorem 4.2. However, Algorithm 1 now runs with a reduced budget $\tilde{B} < B$. We thus need to show that even with a reduced budget, we can still get a good fraction of OPT as long as $\tilde{B}$ is sufficiently large.

**Lemma 4.7.** *Let $\widetilde{\text{OPT}}_b = \max_{S \subseteq V : |S| \leq b} \min_{c \in \tilde{C}} \tilde{f}_c(S)$ for $b \geq 0$. Let $\gamma > 0$ be such that $\tilde{f}_c(v \mid \emptyset) \leq \gamma \widetilde{\text{OPT}}_B$ for all $v \in V$ and $c \in \tilde{C}$. Then, for $\tilde{B} \leq B$,*

$$\widetilde{\text{OPT}}_{\tilde{B}} \geq \left(1 - \sqrt{3\gamma \frac{B}{\tilde{B}} \log k}\right) \frac{\tilde{B}}{B} \widetilde{\text{OPT}}_B.$$

*Proof.* We use a probabilistic argument. As before, we abuse notation and use $\widetilde{\text{OPT}}_b$ to denote both the optimum solution and its value. Let $x^* \in \{0, 1\}^V$ be such that $x_v^* = 1$ if $v \in \widetilde{\text{OPT}}_B$ and $x_v^* = 0$ otherwise. We define $\tilde{x} = (1 - \frac{\tilde{B}}{B}) x \in [0, 1]^V$ as a similar vector but with a reduced budget. Let $\tilde{F}_c \colon [0, 1]^V \to \mathbb{R}$ be the multilinear extension of $\tilde{f}_c$ for each $c \in \tilde{C}$. Since $x^* \in \{0, 1\}^V$ we have $\tilde{f}_c(\widetilde{\text{OPT}}_B) = \tilde{F}_c(x^*)$. As argued in Lemma 3 of Udwani's work (Udwani, 2018), we have

$$\tilde{F}_c(\tilde{x}) \geq \frac{\tilde{B}}{B} \tilde{F}_c(x^*) = \frac{\tilde{B}}{B} \tilde{f}_c(\widetilde{\text{OPT}}_B), \tag{5}$$

which is due to Jensen's inequality and since the multilinear extension is concave in positive direction. We now use swap rounding (Chekuri et al., 2010) to obtain a set $\tilde{S} \subseteq V$ from $\tilde{x}$ of size $|\tilde{S}| = \tilde{B}$. Importantly, swap rounding is an oblivious rounding scheme, meaning it does not evaluate $\tilde{f}_c$. As such, we do not create a rounded solution specific to $\tilde{f}_c$, but the

guarantee on the rounded solution holds for all $c \in \tilde{C}$. We have the following guarantee (Chekuri et al., 2010) for a rounded solution $\tilde{S}$, for all $\delta > 0$,

$$\Pr[\tilde{f}_c(\tilde{S}) \leq (1 - \delta)\tilde{F}_c(\tilde{x})] \leq \exp\left(-\frac{\tilde{F}_c(\tilde{x})\delta^2}{2M_c}\right) \leq \exp\left(-\frac{\tilde{B}\tilde{f}_c(\widetilde{\mathrm{OPT}}_B)\delta^2}{2B\gamma\widetilde{\mathrm{OPT}}_B}\right) \leq \exp\left(-\frac{\delta^2\tilde{B}}{2\gamma B}\right). \tag{6}$$

For the second inequality, we used that $M_c = \max_{v \in V} \tilde{f}_c(v \mid \emptyset) \leq \gamma\widetilde{\mathrm{OPT}}_B$ and Inequality (5). For the last inequality, we used that $\tilde{f}_c(\widetilde{\mathrm{OPT}}_B) \geq \widetilde{\mathrm{OPT}}_B$. We need that the RHS of Inequality (6) is less than $1/k$ to guarantee that the rounded solution exceeds $\tilde{F}_c(\tilde{x})$ for all $c \in \tilde{C}$. In particular, we choose

$$\delta = \sqrt{3\gamma\frac{B}{\tilde{B}}\log k} \implies \exp\left(-\frac{\delta^2\tilde{B}}{2\gamma B}\right) < \frac{1}{k}.$$

By a union bound over all $c \in C$,

$$\Pr\left[\text{there exists a } c \in \tilde{C} \text{ with } \tilde{f}_c(\tilde{S}) \leq (1 - \delta)\tilde{F}_c(\tilde{x})\right] < k \cdot \frac{1}{k} = 1.$$

As such, the event that $\tilde{S}$ obtains value $\tilde{f}_c(\tilde{S}) \geq (1 - \delta)\tilde{F}_c(\tilde{x})$ for all $c \in C$ simultaneously has non-zero probability, meaning that such a set $\tilde{S}$ exists. Let $\tilde{S}$ now be this set. We have for all $c \in \tilde{C}$ that

$$\widetilde{\mathrm{OPT}}_{\tilde{B}} \geq \tilde{f}_c(\tilde{S}) \geq (1 - \delta)\tilde{F}_c(\tilde{x}) \geq (1 - \delta)\frac{\tilde{B}}{B}\widetilde{\mathrm{OPT}}_B.$$

$\square$

We run the pre-processing Algorithm 3 for a value $B > 0$ which we will specify later in Theorem 4.8. This outputs a set of colors $\tilde{C} \subseteq C$ and a partial solution $T \subseteq V$, and we run Algorithm 2 use objective functions defined as $\tilde{f}_c(A) = f_c(A \cup T)$ for $c \in \tilde{C}$ and the budget $\tilde{B} = B - |T|$. Since for $c \in C \setminus \tilde{C}$ we have already reached the optimum value, we simply ignore those colors and do not pass them to Algorithm 2. Algorithm 2 outputs a set $\tilde{S} \subseteq V$ and the final solution of the combined algorithm is $T \cup \tilde{S}$.

We can now put everything together to get the guarantee on the combined algorithm that only requires a large budget.

**Theorem 4.8.** *If $B \geq 108\frac{k}{\epsilon^3}\log k$ then*

$$\min_{c \in C} f_c(T \cup \tilde{S}) \geq \left(1 - \frac{1}{e} - \epsilon\right)\mathrm{OPT}$$

*with probability at least $1 - \delta$.*

*Proof.* We use $B = \frac{1}{\gamma}$ where $\gamma = \frac{\epsilon^2}{36\log k}$ for Algorithm 3 which yields a partial solution $T$ and colors $\tilde{C} \subseteq C$. We show the theorem statement for all $c \in C$ separately. If $c \notin \tilde{C}$, we have by monotonicity that

$$f_c(T \cup \tilde{S}) \geq f_c(T) \geq \mathrm{OPT}.$$

Let now $c \in \tilde{C}$. After the pre-processing, we have by Lemma A.1 that $|T| \leq \frac{k}{\gamma}$ so $\tilde{B} = B - |T| \geq B - \frac{k}{\gamma}$ and hence, by Lemma 4.7,

$$\widetilde{\mathrm{OPT}}_{\tilde{B}} \geq \widetilde{\mathrm{OPT}}_B\left(1 - \frac{k}{B\gamma}\right)\left(1 - \sqrt{3\gamma\frac{1}{1 - \frac{k}{B\gamma}}\log k}\right). \tag{7}$$

Now, since $B \geq \frac{3k}{\epsilon\gamma}$ we get $\frac{k}{B\gamma} \leq \frac{\epsilon}{3}$ as well as

$$3\gamma\frac{1}{1 - \frac{k}{B\gamma}}\log k \leq 3\gamma\frac{1}{1 - \frac{\epsilon}{3}}\log k < 4\gamma\log k = \frac{\epsilon^2}{9},$$

---

**Algorithm 4:** Solving the LP with MWU and Lazy Evaluations

---

**Input:** Monotone and submodular set functions $f_c \colon 2^V \to \mathbb{R}$ for $c \in C$, partial solution $S \subseteq V$, upper bounds $g_c(v)$ for all $v \in V$ and $c \in C$ such that $g_c(v) \geq f_c(v \mid S)$, step size $\eta > 0$, and number of iterations $T$

1  Initialize $y_c \leftarrow 1/|C|$ for all $c \in C$ and $\overline{x}_v \leftarrow 0$ for all $v \in V$;

2  **for** $t = 1, 2, \ldots, T$ **do**

3      Let $v^* \leftarrow \perp$;

4      **for** $v \in V$ in order decreasing in $\sum_{c \in C} y_c g_c(v)$ **do**

5         **if** $v^* \neq \perp$ and $\sum_{c \in C} y_c g_c(v) \leq \sum_{c \in C} y_c g_c(v^*)$ **then**

6            **break**;

7         If not yet computed, evaluate $f_c(v \mid S)$ for all $c \in C$;

8         Update $g_c(v) \leftarrow f_c(v \mid S)$ for all $c \in C$;

9         **if** $v^* = \perp$ or $\sum_{c \in C} y_c g_c(v) > \sum_{c \in C} y_c g_c(v^*)$ **then**

10           Set $v^* \leftarrow v$;

11     Update $y_c \leftarrow y_c(1 - \eta \ell_c(v^*)) = y_c(1 - \eta(B f_c(v^* \mid S) + f_c(S)))$;

12     Normalize $y_c \leftarrow y_c / \|y\|_1$ ;

13     Add $\overline{x}_{v^*} \leftarrow \overline{x}_{v^*} + \frac{1}{T}$;

14 **return** $\overline{x}$ and the updated upper bounds $g_c(v)$ for all $v \in V$ and $c \in C$.

---

which means we can bound (7) further and obtain

$$\widetilde{\mathrm{OPT}}_{\tilde{B}} \geq \left(1 - \frac{2}{3}\epsilon\right) \widetilde{\mathrm{OPT}}_B. \tag{8}$$

Lemma A.1 also says that after pre-processing, the maximum singleton marginal gain is

$$\tilde{M}_c = \max_{v \in V} \tilde{f}_c(v \mid \emptyset) \leq \gamma \mathrm{OPT} \leq \gamma \widetilde{\mathrm{OPT}}_B \leq 2\gamma \widetilde{\mathrm{OPT}}_{\tilde{B}},$$

where the last inequality follows from (8). Hence, by Theorem 4.2,

$$\tilde{f}_c(S) \geq \left(1 - \frac{1}{e} - \frac{\epsilon}{3}\right) \widetilde{\mathrm{OPT}}_{\tilde{B}}. \tag{9}$$

with probability at least $1 - \delta$. Let us condition on the case that Algorithm 2 was successful and (9) holds. In this case, putting everything together yields

$$
\begin{aligned}
f_c(T \cup \tilde{S}) = \tilde{f}_c(\tilde{S}) && \text{(definition of } \tilde{f}_c) \\
\geq \left(1 - \frac{1}{e} - \frac{\epsilon}{3}\right) \widetilde{\mathrm{OPT}}_{\tilde{B}} && \text{(by (9))} \\
\geq \left(1 - \frac{1}{e} - \frac{\epsilon}{3}\right)\left(1 - \frac{2}{3}\epsilon\right) \widetilde{\mathrm{OPT}}_B && \text{(by (8))} \\
\geq \left(1 - \frac{1}{e} - \epsilon\right) \widetilde{\mathrm{OPT}}_B && \\
\geq \left(1 - \frac{1}{e} - \epsilon\right) \mathrm{OPT}. && \text{(monotonicity)}
\end{aligned}
$$

$\square$

## A.3. Missing Details and Proofs from Section 4.3

Let us now describe and motivate the design of Algorithm 4. Imagine a single iteration of Algorithm 1 where we have a partial solution $S = S^{(i-1)}$. We can formulate the LP (3) as a zero-sum game with a payoff for each element $v \in V$ and each color $c \in C$ defined as

$$\ell_c(v) = B f_c(v \mid S) + f_c(S). \tag{10}$$

We write our LP in terms of this payoff:

$$\text{LP}(S) = \max_{x \in \Delta_V} \min_{c \in C} \left\{ \sum_{v \in V} x_v \ell_c(v) \right\} = \min_{y \in \Delta_C} \max_{v \in V} \left\{ \sum_{c \in C} y_c \ell_c(v) \right\}$$

where we are able to exchange minimum and maximum as this presents a zero-sum game (v. Neumann, 1928). We use the min-max formulation where the first player plays a distribution $y \in \Delta_C$ over colors and the second player responds with the best response

$$v^* = \arg\max_{v \in V} \sum_{c \in C} y_c \ell_c(v) = \arg\max_{v \in V} \sum_{c \in C} y_c f_c(v \mid S)$$

where the equality is true since $f_c(S)$ is constant in $v$. Since the RHS is just a linear combination of marginal gains, we can now employ lazy evaluations when finding the maximizer $v^*$: While maintaining upper bounds $g_c(v) \geq f_c(v \mid S)$ for all $v \in V, c \in C$, it verify that $\sum_{c \in C} y_c f_c(v \mid S) \geq \sum_{c \in C} y_c g_c(v)$. Since the solution $S$ we build increases and marginal gains are decreasing due to submodularity, we use prior marginal gains for $g_c(v)$, and update the upper bounds until we can show optimality of $v^*$.

Finally, we understand the colors as experts and, treating $\ell(v) = (\ell_c(v))_{c \in C} \in \mathbb{R}^C$ as a loss vector, we update $y$ via multiplicative weight updates.

We now want to bound the required number of iterations $T$. The MWU can only guarantee an approximate solution to the LP with a distribution $\overline{x} \in \Delta_V$, but this is sufficient for our purposes. Indeed, the following holds:

**Lemma A.2.** *Assume that in every iteration $1 \leq i \leq B$ of Algorithm 1, we use an approximate solution $\overline{x}^{(i)}$ with*

$$\sum_{v \in V} \overline{x}_v^{(i)} f_c(v \mid S^{(i)}) \geq \frac{1}{B}((1 - \epsilon)\text{OPT} - f_c(S^{(i)}))$$

*for all $v \in V$ and $S \subseteq V$. Then, Algorithm 1 still outputs a solution $S$ such that $f_c(S) \geq (1 - \frac{1}{e} - O(\epsilon))\text{OPT}$ for all colors $c \in C$, under the same conditions as Theorem 4.2.*

*Proof.* The proof is simple and follows, for example, by replacing OPT by $(1 - \epsilon)\text{OPT}$ in the proof of Theorem 4.2 and related lemmas. $\square$

We therefore need to show that using sufficiently many iterations, the multiplicative weights update can obtain an approximation $\overline{x}$ that satisfies the condition of Lemma A.2.

We use the typical regret guarantee (for example, see Arora et al. (2012)) for MWU, which states that

$$\min_{y \in \Delta_C} \sum_{t=1}^T \langle \ell(v^{(t)}), y \rangle \geq \sum_{t=1}^T \langle \ell(v^{(t)}), y^{(t)} \rangle - \frac{\log k}{\eta} - \eta \sum_{t=1}^T \|\ell(v^{(t)})\|_\infty^2 \tag{11}$$

where $y^{(t)}$ is the weight vector $y$ at the beginning of iteration $t$, $\ell(v) = (\ell_c(v))_{c \in C} \in \mathbb{R}^C$ the loss vector as defined in (10), and $v^{(t)}$ the best response to $y^{(t)}$.

**Lemma A.3.** *Using $T = 16B^2 M^2 \frac{\log k}{\epsilon^2 \text{OPT}^2}$ and an appropriate choice of $\eta > 0$, the solution $\overline{x} \in \Delta_V$ is such that for all sets $S \subseteq V$ with $|S| \leq B$,*

$$\sum_{v \in V} \overline{x}_v f_c(v \mid S) \geq \frac{1}{B}((1 - \epsilon)\text{OPT} - f_c(S)).$$

*Proof.* Let $\overline{y} = \frac{1}{T} \sum_{t=1}^T y^{(t)} \in \Delta_C$. Note first that

$$\frac{1}{T} \sum_{t=1}^T \langle \ell(v^{(t)}), y^{(t)} \rangle = \frac{1}{T} \sum_{t=1}^T \max_{v \in V} \langle \ell(v), y^{(t)} \rangle \geq \max_{v \in V} \frac{1}{T} \sum_{t=1}^T \langle \ell(v), y^{(t)} \rangle = \max_{v \in V} \langle \ell(v), \overline{y} \rangle \geq \text{OPT}.$$

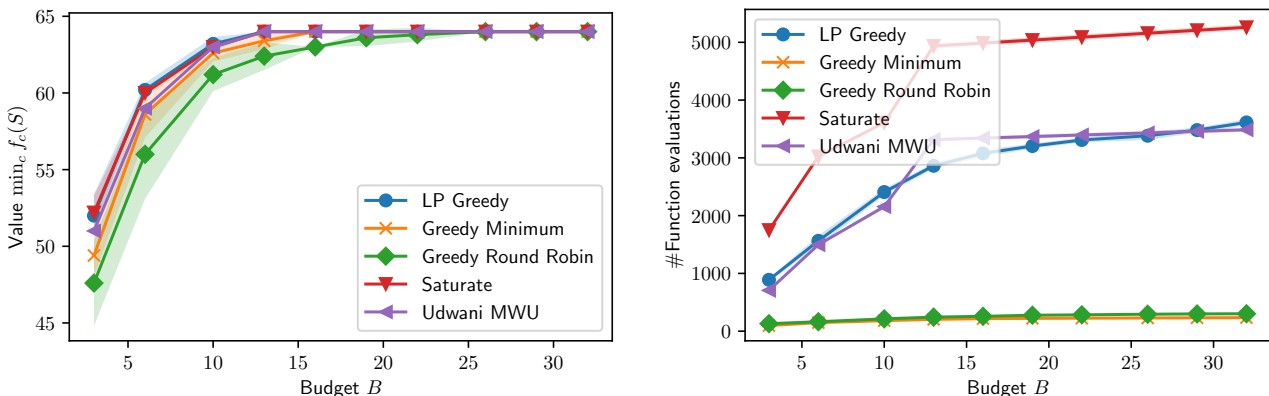

*Figure 5.* Multiobjective submodular maximization for max-$k$-cover. We use $k = 20$ Barabási-Albert graphs on $n = 64$ nodes. We show the function value (top) and the number of evaluations (bottom). We report mean and standard deviation over 5 runs.

As such, the regret guarantee (11) implies

$$\min_{c \in C} \frac{1}{T} \sum_{t=1}^{T} \ell_c(v^{(t)}) = \min_{y \in \Delta_C} \frac{1}{T} \sum_{t=1}^{T} \langle \ell(v^{(t)}), y \rangle \geq \text{OPT} - \frac{\log k}{T\eta} - \frac{\eta}{T} \sum_{t=1}^{T} \| \ell(v^{(t)}) \|_\infty^2. \tag{12}$$

Since $\ell_c(v^{(t)}) = B f_c(v \mid S) + f_c(S) \leq BM + \text{OPT} \leq 2BM$ it suffices to set

$$\eta = \frac{\epsilon}{8B^2M} \leq \frac{\epsilon \text{OPT}}{8B^2M^2} \quad \text{and} \quad T = 16 \frac{\log k B^2}{\epsilon^2} \geq 16 \frac{B^2 M^2 \log k}{\epsilon^2 \text{OPT}^2}$$

so we can further bound (12) by $(1 - \epsilon)\text{OPT}$. Therefore, we have for all $c \in C$ that

$$\sum_{v \in V} \overline{x}_v (B f_c(v \mid S) + f(S)) = \frac{1}{T} \sum_{t=1}^{T} \ell_c(v^{(t)}) \geq (1 - \epsilon)\text{OPT} = \frac{\log k}{\epsilon \eta \text{OPT}},$$

where the equality is by definition of $\overline{x}$ and the loss $\ell(v)$. Since $\overline{x} \in \Delta_V$, this is equivalent to the statement we wanted to show, so we are done. $\qquad \square$

**Theorem 4.9.** *Our algorithm for multiobjective submodular maximization runs in time $O(nB^3 \frac{1}{\epsilon^2} k \log(k) \log(1/\delta))$ and requires $O(nBk \log(1/\delta))$ function evaluations.*

*Proof.* The pre-processing takes time $O(kn)$ since it has at most one function evaluation per color $c \in C$ and element $v \in V$. Algorithm 2 runs through a total of $O(B \log(1/\delta))$ iterations: $\log(2/\delta)$ iterations of the outer loop and $B$ iterations of the inner loop. To solve the LP in each iteration, we need the marginal gains for all colors $c \in C$ and elements $v \in V$, which thus requires $O(kn)$ function evaluations. We can solve the LP approximately which, as shown in Lemma A.3, requires $T = O(B^2 M \frac{\log k}{\epsilon^2 \text{OPT}}) = O(B^2 \frac{1}{\epsilon^2} \log k)$ iterations. Each iteration of the multiplicative weights update involves at most $nk$ function evaluations. Overall, we thus have a running time of $O(nB^3 \frac{1}{\epsilon^2} k \log(k) \log(1/\delta))$ and $O(nBk)$ function evaluations. $\qquad \square$

## B. Further Experimental Results

### B.1. Max-$k$-Coverage

Figures 5 and 6 show our results for Barabási-Albert and Erdős-Rényi random graphs, respectively. We obtain a preferential-attachment graph in the Barabási-Albert model by iteratively connecting each node to $d = 5$ existing nodes, with probability

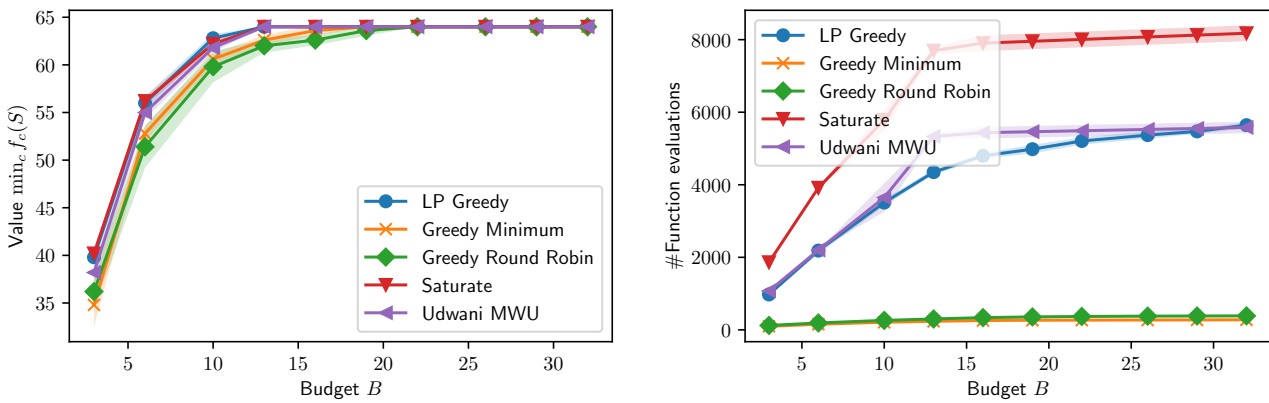

*Figure 6.* Multiobjective submodular maximization for max-$k$-cover. We use $k = 20$ Erdős-Rényi graphs on $n = 64$ nodes. We show the function value (top) and the number of evaluations (bottom). We report mean and standard deviation over 5 runs.

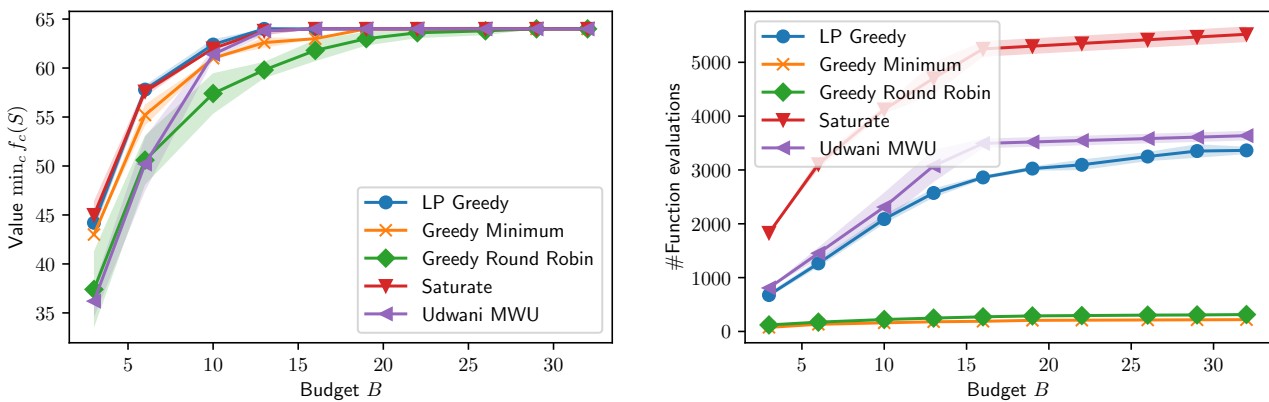

*Figure 7.* Multiobjective submodular maximization for max-$k$-cover. We use $k = 20$ Barabási-Albert graphs on $n = 64$ nodes for varying $d$. We show the function value (top) and the number of evaluations (bottom). We report mean and standard deviation over 5 runs.

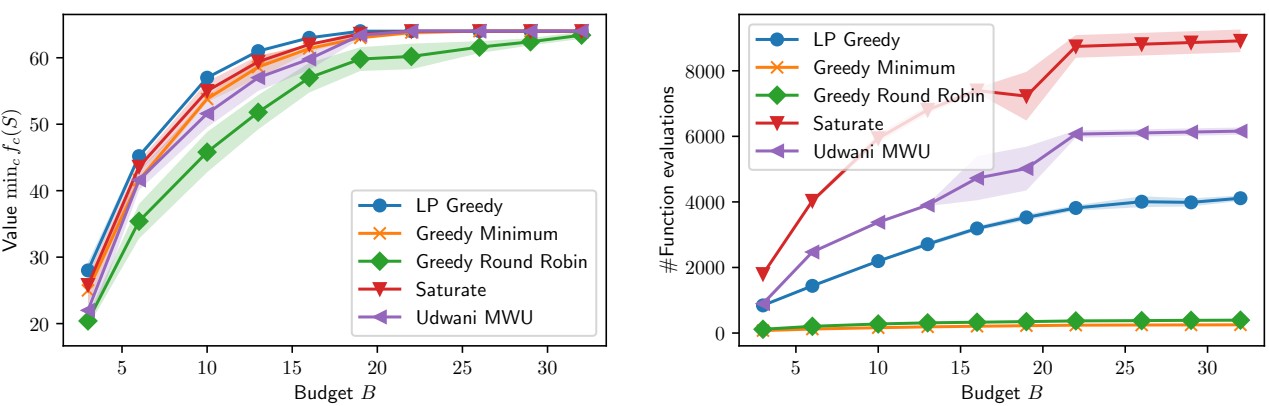

*Figure 8.* Multiobjective submodular maximization for max-$k$-cover. We use $k = 20$ Erdős-Rényi graphs on $n = 64$ nodes for varying $p$. We show the function value (top) and the number of evaluations (bottom). We report mean and standard deviation over 5 runs.

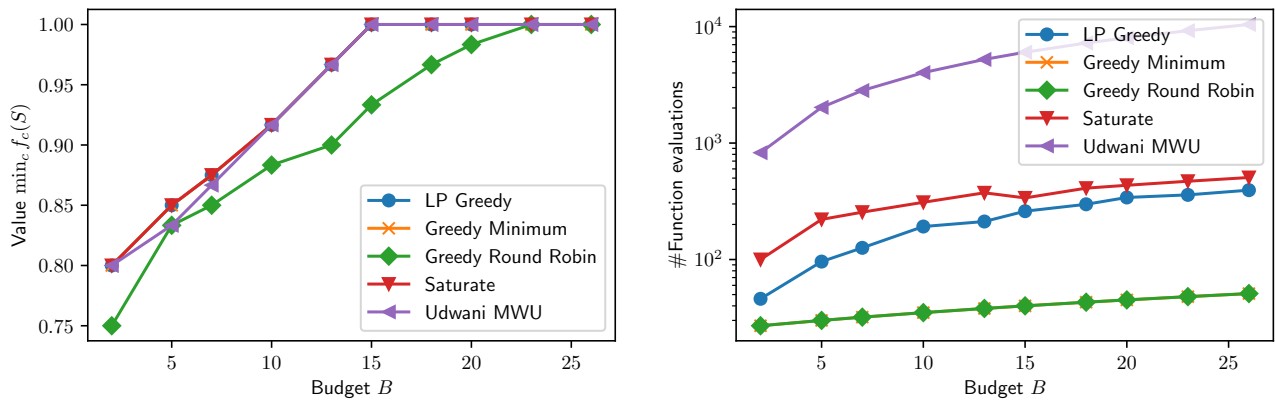

*Figure 9.* Fair centrality maximization on the Amazon co-purchasing graph *Movies & TV* with $n = 23$ nodes and $k = 2$ colors.

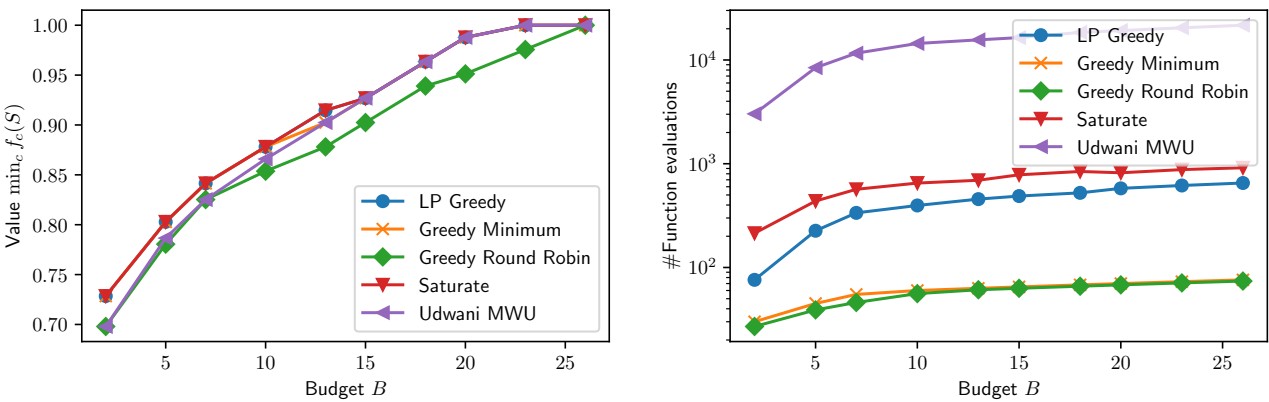

*Figure 10.* Fair centrality maximization on the Amazon co-purchasing graph *Musical Instruments* with $n = 46$ nodes and $k = 2$ colors.

proportional to their degrees. We obtain a random graph in the Erdős-Rényi model by including each pair as an edge with probability $0.1$.

Additionally, we create more difficult synthetic instances where the graph properties differ per color. For Erdős-Rényi random graphs, we create an instance where we use $p_c = 0.1 + \frac{c}{50} \in [0.1, 0.5]$ for colors $1 \leq c \leq 20$ to generate the $c$-th graph. For Barabási-Albert graphs, we vary the number of $d_c = \lceil 5 + \frac{c}{2} \rceil$ for colors $1 \leq c \leq 20$. Our results in Figures 7 and 8 show that the our algorithm performs better under imbalance compared to heuristics such as SATURATE.

### B.2. Fair Centrality Maximization

Figures 9 through 25 show omitted results on Amazon co-purchasing graphs.

### B.3. Fair Influence Maximization

Figures 26 through 29 show omitted results on an Antelope Valley network for four remaining attributes.

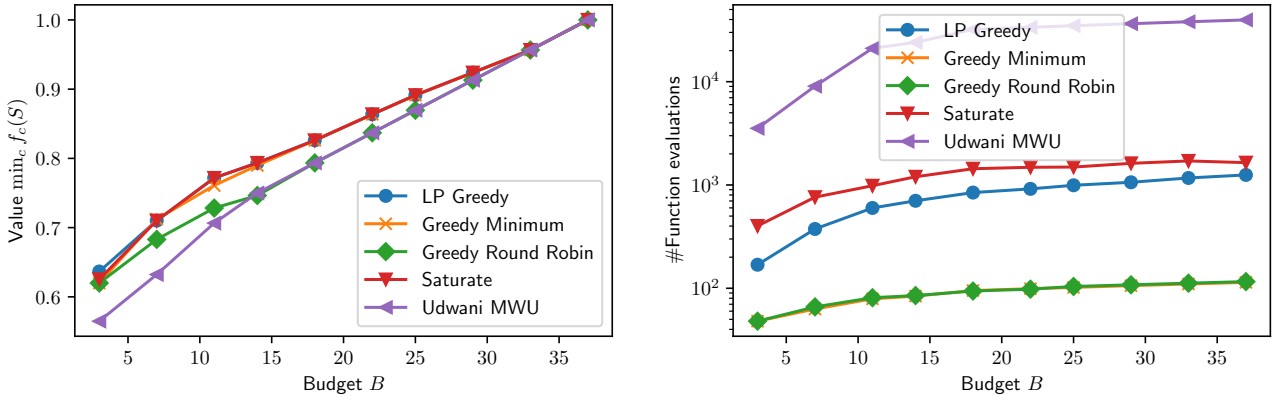

*Figure 11.* Fair centrality maximization on the Amazon co-purchasing graph *All Electronics* with $n = 47$ nodes and $k = 2$ colors.

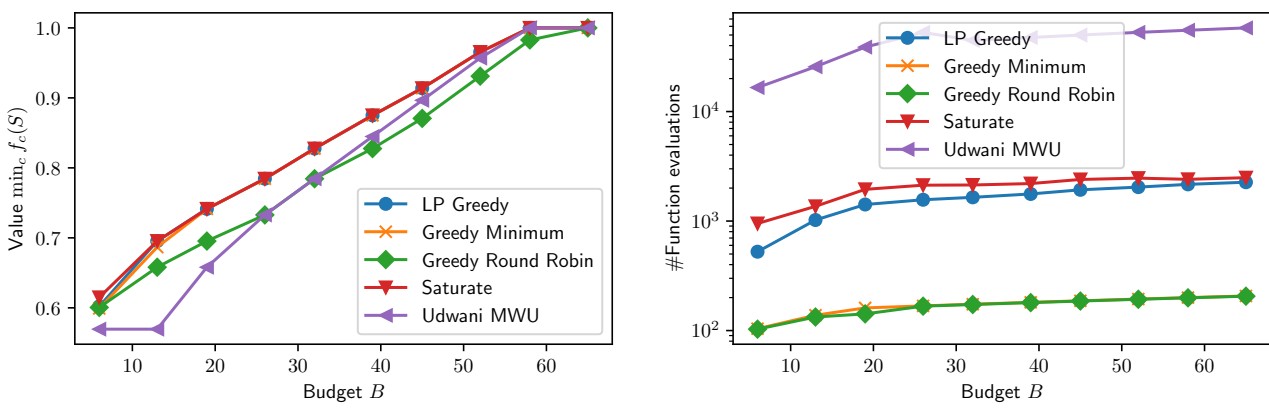

*Figure 12.* Fair centrality maximization on the Amazon co-purchasing graph *Computers* with $n = 59$ nodes and $k = 2$ colors.

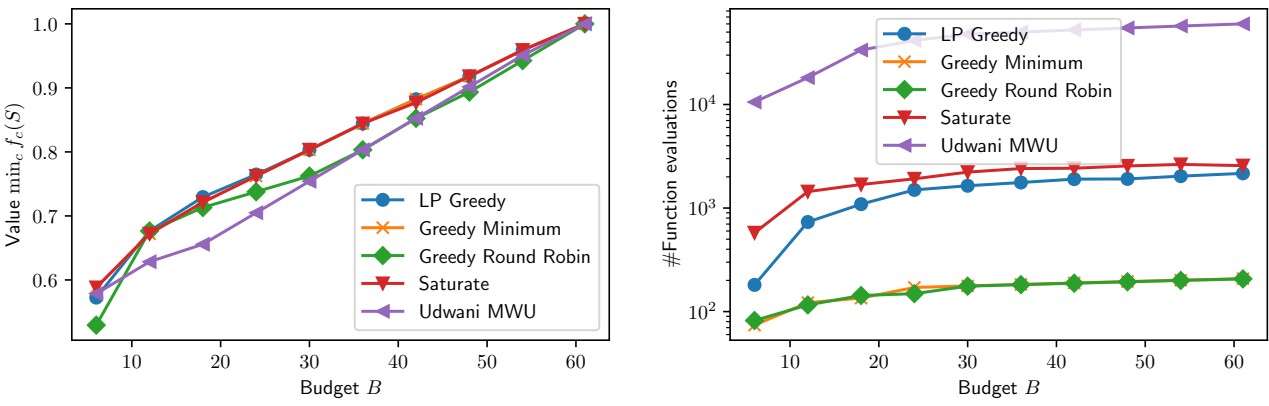

*Figure 13.* Fair centrality maximization on the Amazon co-purchasing graph *Home Audio & Theater* with $n = 77$ nodes and $k = 2$ colors.

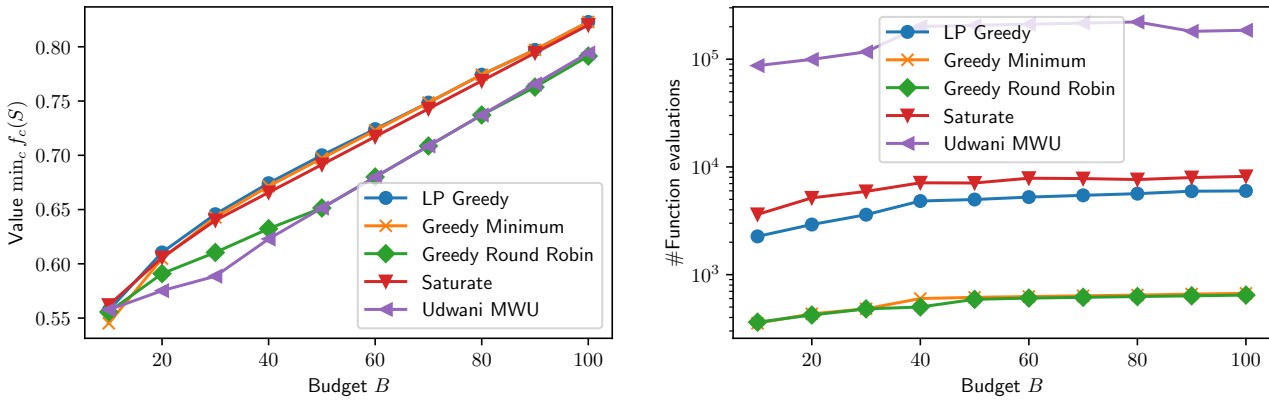

*Figure 14.* Fair centrality maximization on the Amazon co-purchasing graph *Camera & Photo* with $n = 202$ nodes and $k = 2$ colors.

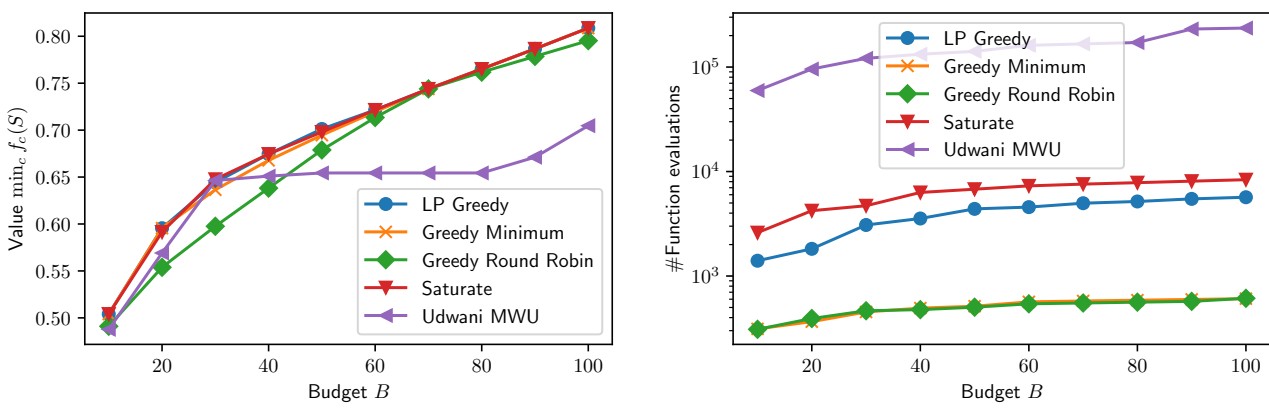

*Figure 15.* Fair centrality maximization on the Amazon co-purchasing graph *Baby* with $n = 228$ nodes and $k = 2$ colors.

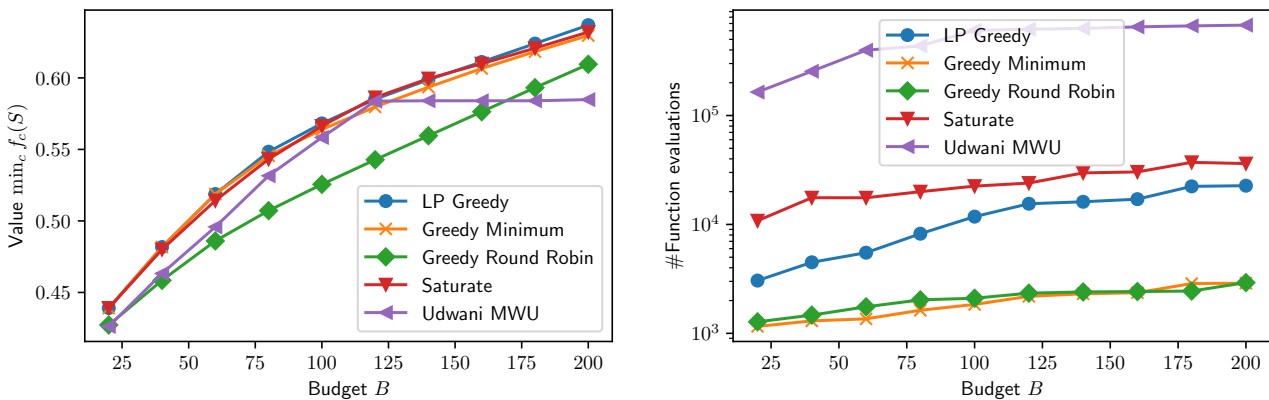

*Figure 16.* Fair centrality maximization on the Amazon co-purchasing graph *Luxury Beauty* with $n = 1037$ nodes and $k = 2$ colors.

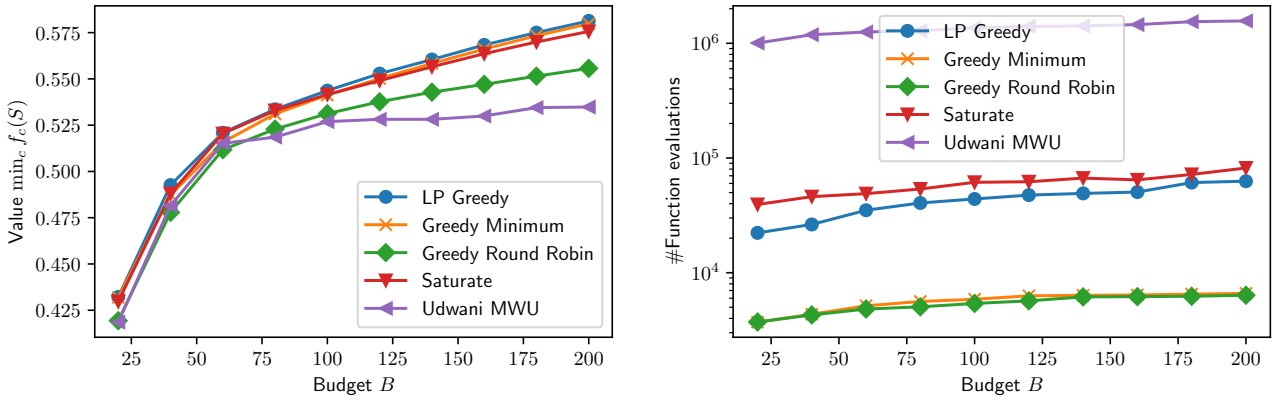

*Figure 17.* Fair centrality maximization on the Amazon co-purchasing graph *Pet Supplies* with $n = 1785$ nodes and $k = 2$ colors.

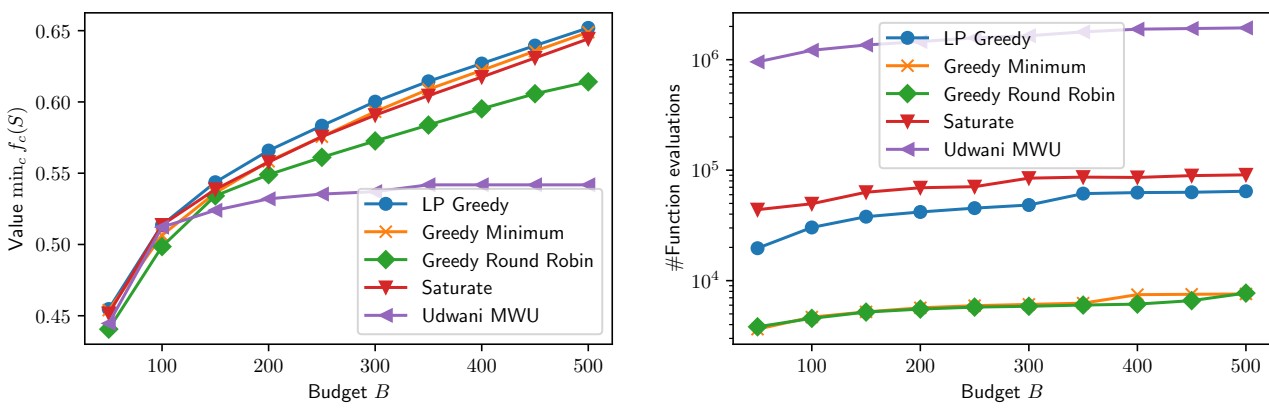

*Figure 18.* Fair centrality maximization on the Amazon co-purchasing graph *Industrial & Scientific* with $n = 2005$ nodes and $k = 2$ colors.

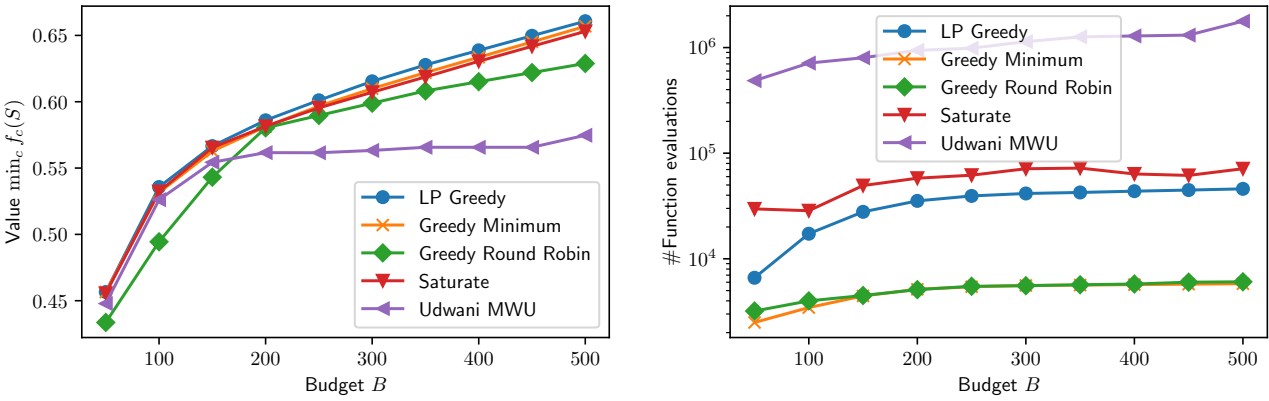

*Figure 19.* Fair centrality maximization on the Amazon co-purchasing graph *Office Products* with $n = 2281$ nodes and $k = 2$ colors.

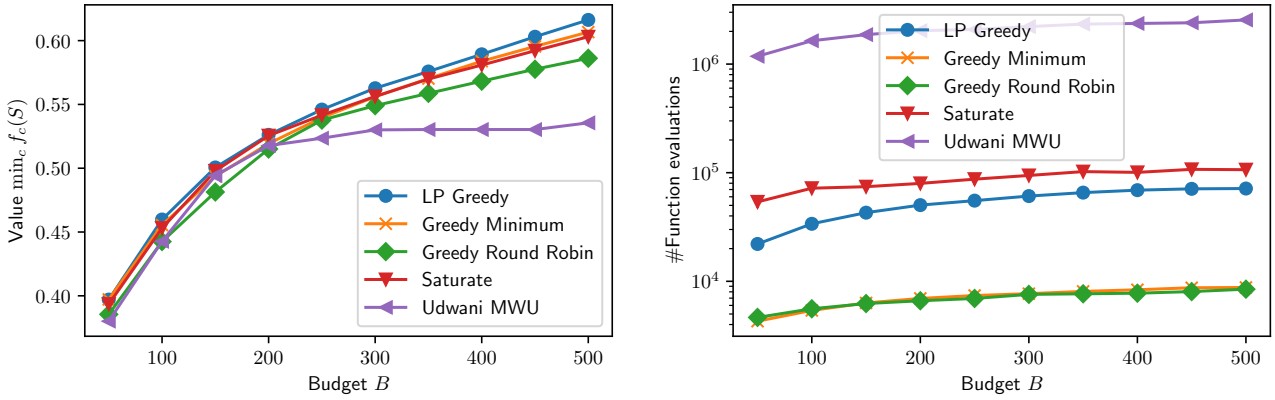

*Figure 20.* Fair centrality maximization on the Amazon co-purchasing graph *Books* with $n = 2495$ nodes and $k = 2$ colors.

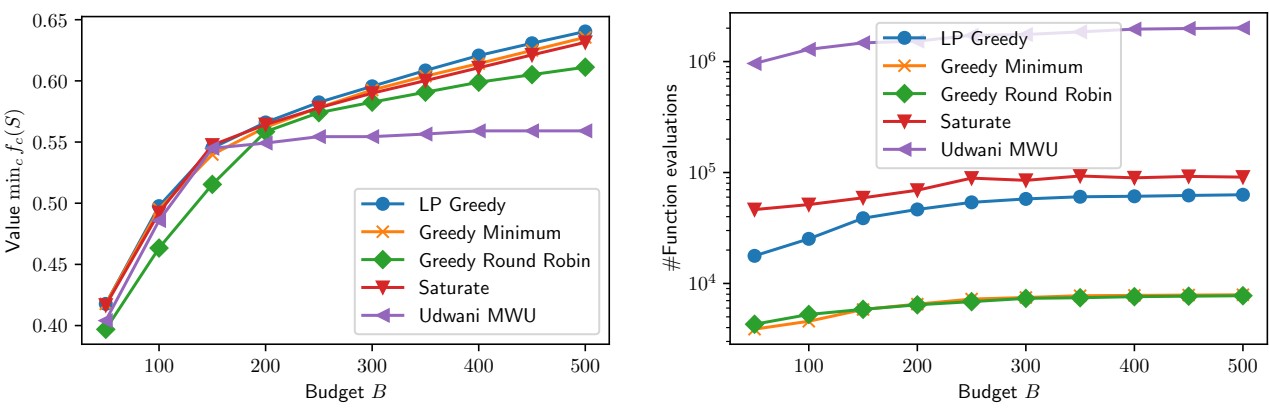

*Figure 21.* Fair centrality maximization on the Amazon co-purchasing graph *Home Improvement* with $n = 2565$ nodes and $k = 2$ colors.

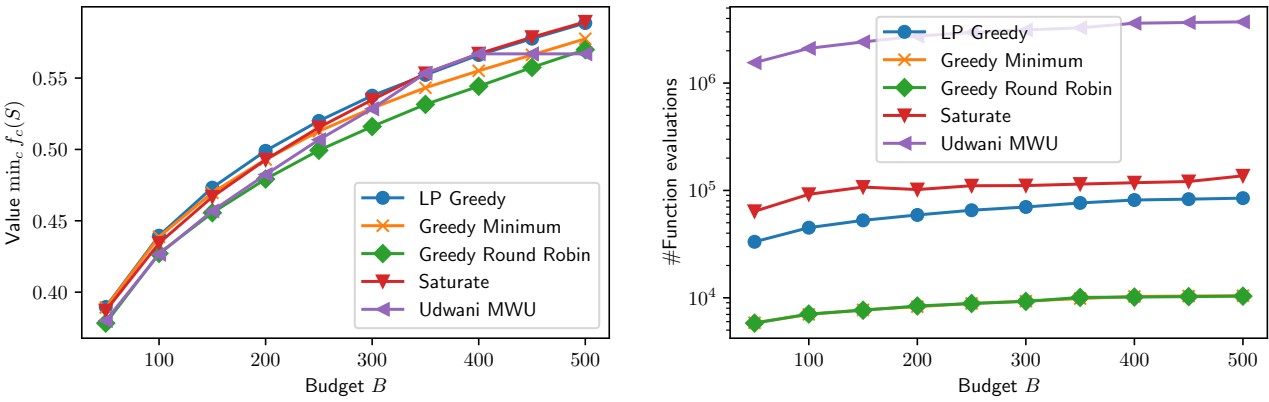

*Figure 22.* Fair centrality maximization on the Amazon co-purchasing graph *Health & Personal Care* with $n = 3010$ nodes and $k = 2$ colors.

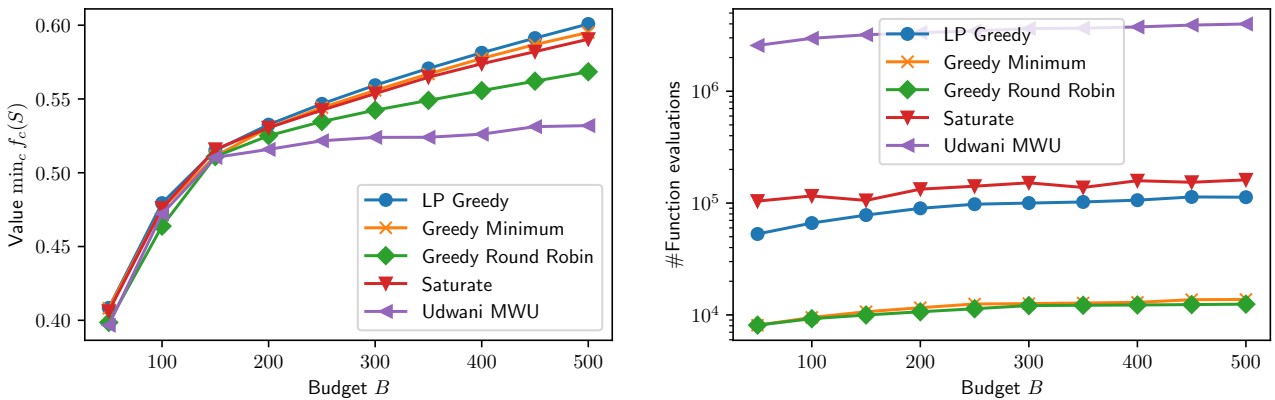

*Figure 23.* Fair centrality maximization on the Amazon co-purchasing graph *Sports & Outdoors* with $n = 3214$ nodes and $k = 2$ colors.

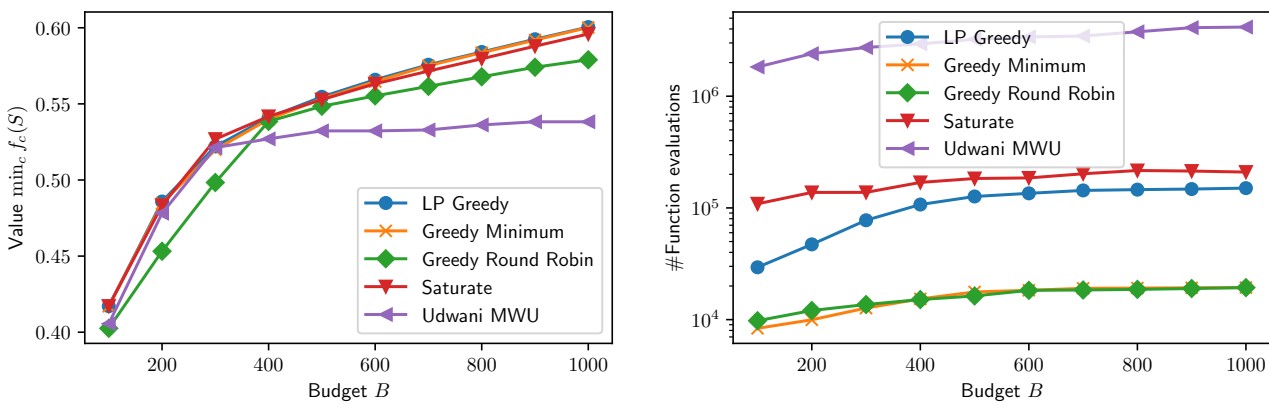

*Figure 24.* Fair centrality maximization on the Amazon co-purchasing graph *Grocery* with $n = 6433$ nodes and $k = 2$ colors.

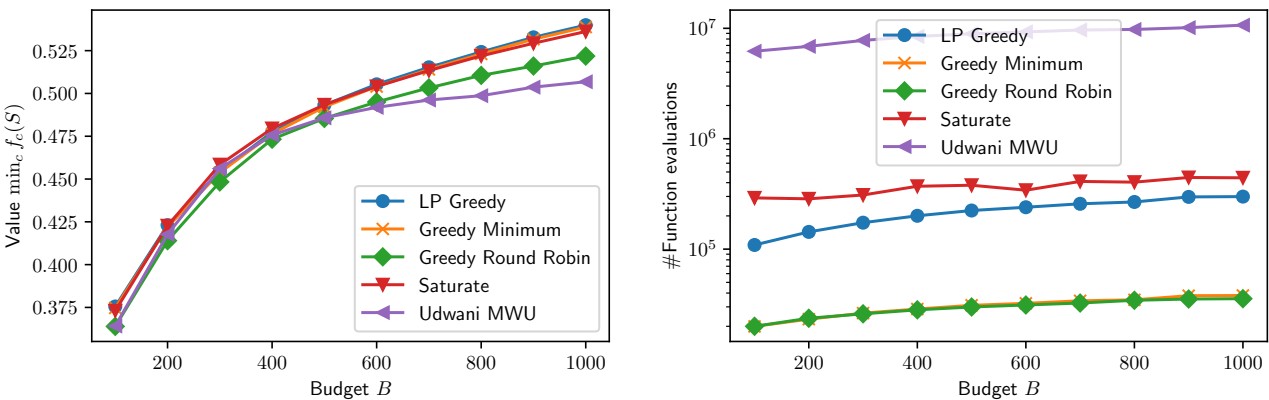

*Figure 25.* Fair centrality maximization on the Amazon co-purchasing graph *Amazon Home* with $n = 10378$ nodes and $k = 2$ colors.

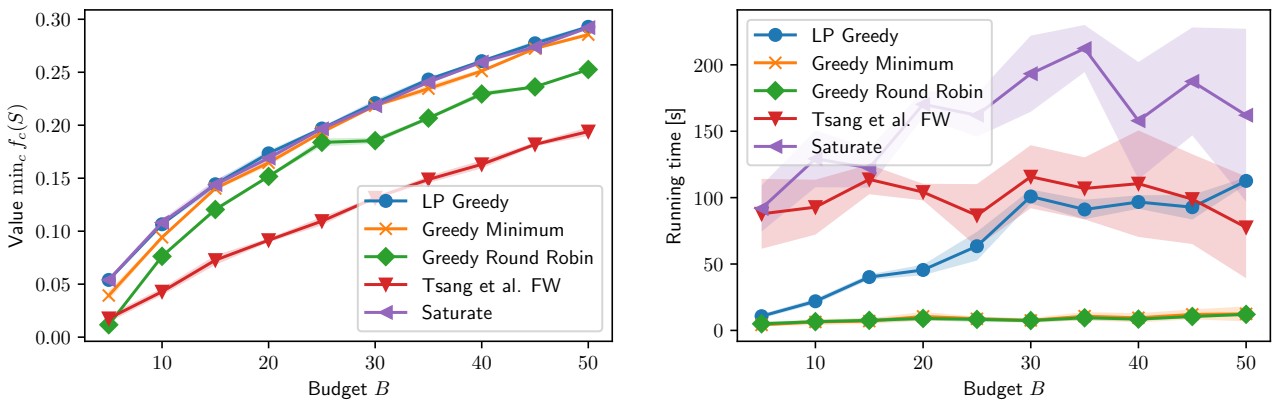

*Figure 26.* Fair influence maximization on a simulated Antelope Valley network with attribute *age* on $k = 7$ colors.

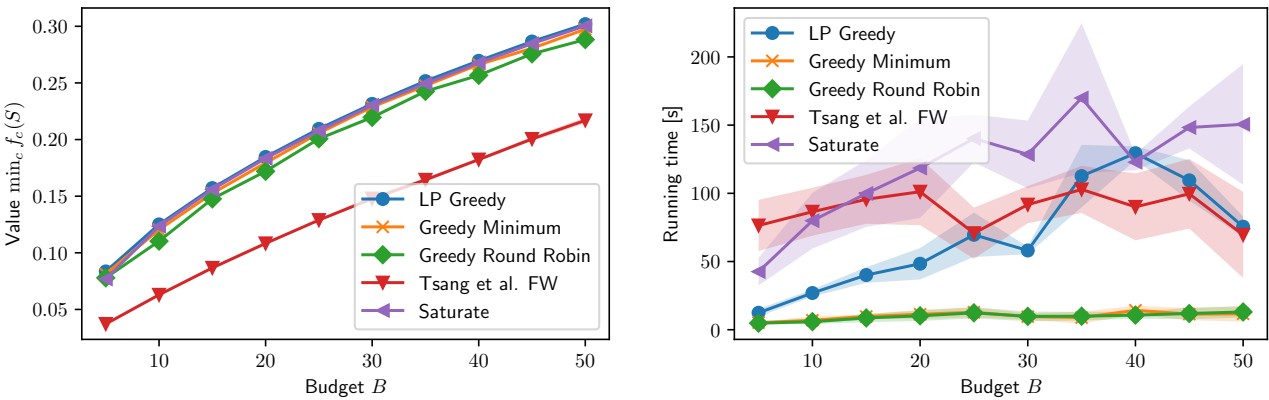

*Figure 27.* Fair influence maximization on a simulated Antelope Valley network with attribute *gender* on $k = 2$ colors.

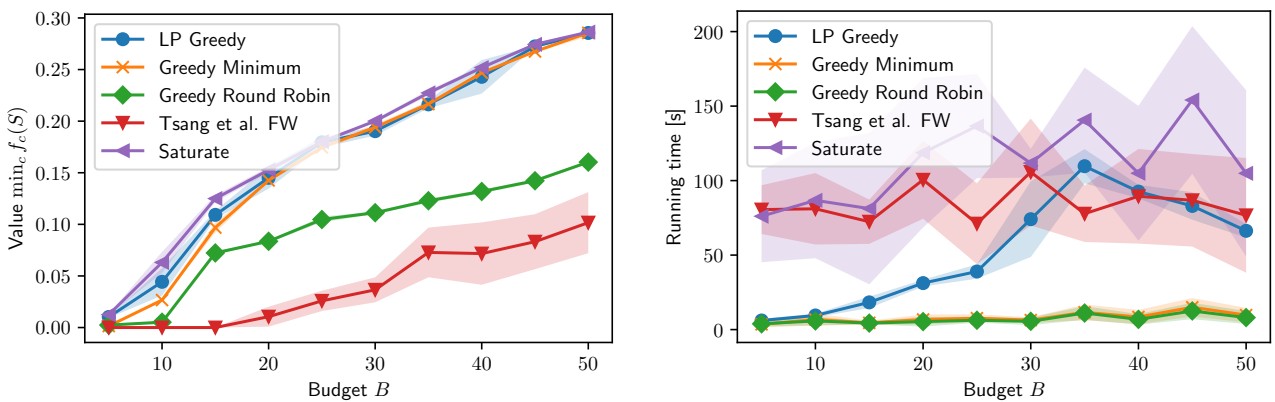

*Figure 28.* Fair influence maximization on a simulated Antelope Valley network with attribute *region* on $k = 13$ colors.

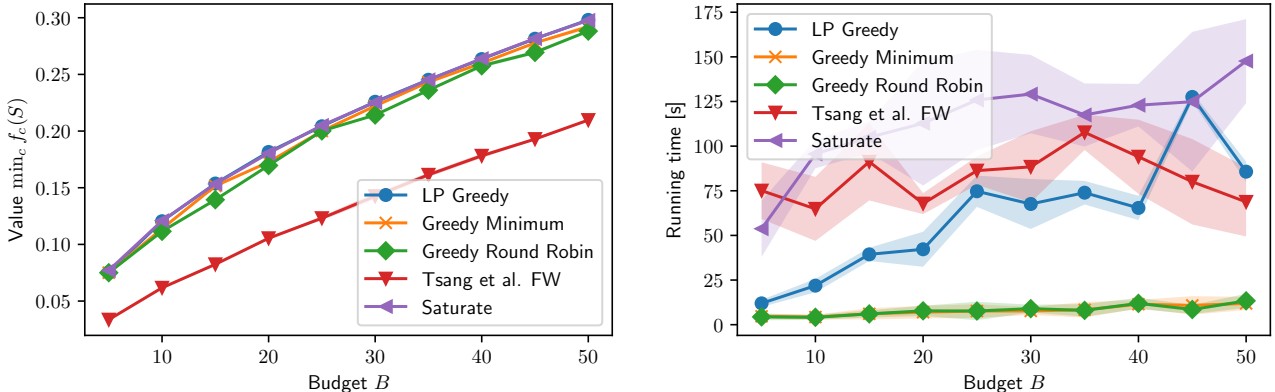

*Figure 29.* Fair influence maximization on a simulated Antelope Valley network with attribute *status* on $k = 3$ colors.

