# OpenReview forum: "An Asymptotically Optimal Approximation Algorithm for Multiobjective Submodular Maximization at Scale"
_ICML.cc/2025/Conference — ICML 2025 poster_

### Official Review · Reviewer_oK8n · 2025-03-07

**Overall Recommendation:** 3

**Summary:**

The paper develops a discrete, scalable algorithm for multiobjective submodular maximization under a cardinality constraint that nearly achieves the optimal (1−1/e) approximation ratio. Instead of relying on continuous relaxations such as the multilinear extension, the proposed approach builds the solution iteratively by solving a linear program at each step to derive a probability distribution over candidate elements, sampling from this distribution, and adding the chosen element to the solution. The algorithm incorporates multiplicative weights updates (MWU) and lazy evaluations to reduce the number of function evaluations, thereby enhancing its efficiency. As an additional contribution, the paper introduces an application to fair centrality maximization, which generalizes standard centrality measures by ensuring fairness across different groups.

**Claims And Evidence:**

The query complexity of O(nBk) appears suboptimal since B approaches n, potentially resulting in quadratic complexity. This does not seem sufficiently efficient for practical applications. The authors should consider developing an approximation with near-linear complexity, which would be more meaningful and applicable in practice.

**Essential References Not Discussed:**

Please refer to the above.

**Experimental Designs Or Analyses:**

The algorithm proposed in the paper demonstrates promising performance in terms of objective function value. However, its runtime is inferior compared to the other two greedy-based algorithms, namely Greedy Minimum and Greedy Round Robin. Additionally, the datasets used in the paper are relatively small, making it difficult to evaluate the efficiency of the algorithms in real-world large-scale environments.

**Methods And Evaluation Criteria:**

While the proposed algorithm is faster than existing methods, its time complexity remains suboptimal for handling the massive data sets encountered in real-world applications. Nevertheless, the paper offers a promising conceptual framework that could inspire further advancements in this area.

**Other Comments Or Suggestions:**

N.A.

**Other Strengths And Weaknesses:**

N.A.

**Questions For Authors:**

Please refer to the above.

**Relation To Broader Scientific Literature:**

The problem addressed in this paper is related to fair submodular optimization, which involves optimizing a submodular function under fairness constraints. However, some important works in this area have been overlooked, such as [1][2].

[1] Fazzone, A., Wang, Y., & Bonchi, F. (2024). Fair Representation in Submodular Subset Selection: A Pareto Optimization Approach. Transactions on Machine Learning Research.

[2] Cui, Shuang, et al. "Fairness in streaming submodular maximization subject to a knapsack constraint." Proceedings of the 30th ACM SIGKDD Conference on Knowledge Discovery and Data Mining. 2024.

**Theoretical Claims:**

I have conducted a preliminary review of the theoretical proofs in the paper, and they appear to be correct. However, I should note that my examination was not exhaustive, and there remains a possibility that some details may have been overlooked.

---

> ### Author Rebuttal · Authors · 2025-03-30
>
> Thank you for your comments and pointing out that our algorithm outperforms existing methods in terms of running time.
>
> > The query complexity of O(nBk) appears suboptimal since B approaches n, potentially resulting in quadratic complexity. This does not seem sufficiently efficient for practical applications. The authors should consider developing an approximation with near-linear complexity, which would be more meaningful and applicable in practice.
>
> > While the proposed algorithm is faster than existing methods, its time complexity remains suboptimal for handling the massive data sets encountered in real-world applications.
>
> A running time of $O(nB)$ is natural for the greedy algorithm which is a standard algorithm in practice and is widely applied on huge datasets. A running time of $O(nBk)$ is therefore quite natural on $k$ objectives. Even in the case where $B=\Omega(n)$, our algorithm still has significantly better running time than all prior works that offer the same approximation guarantee (Chekuri et al. ‘10, the $1-1/e$ algorithm of Udwani ‘18, Tsang et al. ‘19).
>
> As shown in our experiments, our algorithm is vastly more performant than prior work with theoretical guarantees. We are the first to provide an algorithm that scales to ten thousands of nodes (and potentially more; the running time of our algorithm on the largest instances we tried is not yet prohibitive) while providing the best possible approximation ratio.
>
> > However, its runtime is inferior compared to the other two greedy-based algorithms, namely Greedy Minimum and Greedy Round Robin. Additionally, the datasets used in the paper are relatively small, making it difficult to evaluate the efficiency of the algorithms in real-world large-scale environments.
>
> As compared to greedy, our algorithm has theoretical guarantees. We outperform all algorithms with comparable theoretical guarantees in terms of running time, which form a long line of work that improves upon the objective value of the greedy algorithm.
>
> The biggest instance in our experiments has $n=10378$ elements which is substantially larger than any of the instances used in the literature on multiobjective submodular maximization with theoretical guarantees (before us, the largest instance appears in Udwani ‘2018 with $n=1024$ elements).
>
> > However, some important works in this area have been overlooked, such as [1][2].
> > [1] Fazzone, A., Wang, Y., & Bonchi, F. (2024). Fair Representation in Submodular Subset Selection: A Pareto Optimization Approach. Transactions on Machine Learning Research.
> > [2] Cui, Shuang, et al. "Fairness in streaming submodular maximization subject to a knapsack constraint." Proceedings of the 30th ACM SIGKDD Conference on Knowledge Discovery and Data Mining. 2024.
>
> We will cite these papers in our related work along the following discussion:
> - [1] solves a multi-objective problem by extending the saturate algorithm to obtain a sequence of approximately pareto-optimal solutions, while our algorithm gives a single solution for the (max-min) multiobjective problem.
> - [2] considers a fairness notion that enforces constraints on the solution set. (We will add [2] alongside similar works that we cite in our related work section (e.g., Celis et al. ‘18)).

---

> > ### Comment · Reviewer_oK8n · 2025-04-02
> >
> > After reading the rebuttal, I have decided to increase my score.

---

### Official Review · Reviewer_roJW · 2025-03-14

**Overall Recommendation:** 3

**Summary:**

This paper addresses the problem of maximizing the minimum over several submodular functions, known as multiobjective submodular maximization. The authors present the first scalable algorithm that achieves the best-known approximation guarantee. Additionally, they introduce a novel application—fair centrality maximization—and demonstrate how it can be effectively addressed through multiobjective submodular maximization.

**Claims And Evidence:**

+The paper tackles an important problem in the field of submodular optimization, with results that can be applied to solving fairness-aware subset selection problems.
+The paper is well-written, with the algorithm presented in a clear and easy-to-follow manner. The analysis is also clearly outlined. Notably, the algorithm is both scalable and effective, achieving the best approximation ratio without relying on continuous relaxation of the submodular set functions.

However, there are some points of concern:

-The technical novelty of the work seems marginal. While the proposed solution outperforms existing studies in terms of lower running time and improved approximation ratio, the fundamental design and analysis of the algorithm appear to be inspired by previous work, such as Chekuri et al. (2010). Moreover, the results of Chekuri et al. (2010) can be applied to a broader range of constraints, such as matroid constraints. Additionally, I believe their results do not rely on the budget constraint B.

-The paper lacks a direct comparison between their method and the one proposed in Chekuri et al. (2010) in the experimental section. While I understand that Chekuri et al. (2010)’s algorithm may not be practical due to its reliance on multilinear extensions, I would have appreciated seeing some comparative results in a small- or medium-scale setting to better understand the trade-offs.

**Essential References Not Discussed:**

No.

**Experimental Designs Or Analyses:**

Yes.

**Methods And Evaluation Criteria:**

Yes.

**Other Comments Or Suggestions:**

NA.

**Other Strengths And Weaknesses:**

NA.

**Questions For Authors:**

Please refer to my comments in Claims And Evidence.

**Relation To Broader Scientific Literature:**

The paper tackles an important problem in the field of submodular optimization, with results that can be applied to solving fairness-aware subset selection problems.

**Theoretical Claims:**

Yes.

---

> ### Author Rebuttal · Authors · 2025-03-30
>
> Thank you for evaluation and the positive comments on our writing and the scalability of our approach.
>
> > The technical novelty of the work seems marginal. While the proposed solution outperforms existing studies in terms of lower running time and improved approximation ratio, the fundamental design and analysis of the algorithm appear to be inspired by previous work, such as Chekuri et al. (2010). Moreover, the results of Chekuri et al. (2010) can be applied to a broader range of constraints, such as matroid constraints. Additionally, I believe their results do not rely on the budget constraint $B$.
>
> Our method and swap rounding (due to Chekuri et al. (2010) and also used in later works, e.g. in Tsang et al. (2019)) are **different approaches** towards the multiobjective problem. The idea of swap rounding is to build a fractional solution (which they find using gradient ascent over the multilinear extension) and in the last step round it to a discrete solution. Our method adds one (discrete) element in each iteration. This leads to important differences in both the running time and the analysis of both approaches. Since we do not optimize over the multilinear extension, our approach is decisively faster: We only need $O(nBk)$ function evaluations compared to exponentially many queries (and exponential runtime) of Chekuri et al. (2010). We also carry out a different martingale analysis that is tailored to the greedy algorithm. Furthermore, their algorithm also depends on $B$ to establish their approximation guarantee, which is natural for the multiobjective problem to avoid the inapproximability regime ($B<k$) shown by Krause et al. (2008). Another natural reason for a dependence on the budget in swap rounding is that the Chernoff bound obtains better concentration for larger budgets, even for linear functions.
>
> > The paper lacks a direct comparison between their method and the one proposed in Chekuri et al. (2010) in the experimental section. While I understand that Chekuri et al. (2010)’s algorithm may not be practical due to its reliance on multilinear extensions, I would have appreciated seeing some comparative results in a small- or medium-scale setting to better understand the trade-offs.
>
> We compare our paper with the approach of Tsang et al. (2019) which provides a practical implementation of the approach of Chekuri et al. (2010) by performing gradient ascent efficiently via Frank-Wolfe. Their parameters are chosen to make the algorithm practical on special medium-sized instances (their algorithm does not apply to general applications such as fair centrality maximization, which is why we exclude it there). However, their algorithm is still slower and has worse objective value than our algorithm, suggesting that the trade-offs of the unrefined approach of Chekuri et al. (2010) are also poor.

---

### Official Review · Reviewer_vgBn · 2025-03-14

**Overall Recommendation:** 4

**Summary:**

They study the multiobjective monotone submodular maximization under cardinality constraint, where the goal is to select a subset of elements of limited size that maximizes the minimum submodular value among the given functions. Previously, a $(1-\frac{1}{e})$-approximation algorithm was known for this problem using multilinear extension which is impractical. Moreover, there was a practical $(1-\frac{1}{e})^2$-approximation algorithm for this problem. Here they develop a $(1-\frac{1}{e}-\epsilon)$-approximation algorithm which relies on selecting elements step by step. At each step, they try to solve an LP to find a probability distribution which identifies the weight of selecting elements to increase the submodular value enough. This is similar to selecting an element with a marginal gain above a threshold, but in this case, it is more complex as it requires finding a probability distribution via an LP. Finally, they select an element using this probability distribution and add it to their solution.

Since their approach is randomized, they run it multiple times to find a desired solution with higher probability. Note that their algorithm has |B| steps, where |B| is the cardinality constraint. At each step, they run a LP which is time consuming and needs to find marginal value of all elements compared to their current solution. However, they have introduced a lazy update method using multiplicative weights update (MWU) to speed up their algorithm.

Finally, they show the efficiency of their approach with experimental evaluation. In their evaluation, they beat the previous practical algorithms with approximation guarantee. However, it seems a greedy heuristics algorithm finds almost the same submodular value, while having better running time and query complexity. This greedy algorithm tries to select an element that maximizes the submodular value of the function that has the current minimum value.

**Claims And Evidence:**

They proved their lemmas and claims and also ran experiments to show their dominance compared to previous practical works.

**Essential References Not Discussed:**

No.

**Experimental Designs Or Analyses:**

No

**Methods And Evaluation Criteria:**

The methods make sense. They could run experiments on larger instances, especially since the multilinear extension approach despite having the same approximation guarantee is impractical on large instances.

**Other Comments Or Suggestions:**

N/A

**Other Strengths And Weaknesses:**

I like their algorithm, it seems natural and achieves the best known approximation factor for submodular maximization. The paper is very well written and easy to follow and understand. The only weakness is that the greedy minimum algorithm seems to outperform their algorithm in experiments but since that one does not have an approximation guarantee, still their algorithm and its analysis remain interesting.

**Questions For Authors:**

N/A

**Relation To Broader Scientific Literature:**

While there was an algorithm with the same approximation guarantee, it was not practical and the previous practical algorithms had worse approximation factors. They could develop a practical algorithm that matches the best known approximation factor for this problem.

**Theoretical Claims:**

I only checked the proofs in the main part of the paper and not the appendix and they seem fine.

---

> ### Author Rebuttal · Authors · 2025-03-30
>
> Thank you for your comments and the positive feedback on our presentation.
>
> > However, it seems a greedy heuristics algorithm finds almost the same submodular value, while having better running time and query complexity.
>
> > The only weakness is that the greedy minimum algorithm seems to outperform their algorithm in experiments but since that one does not have an approximation guarantee, still their algorithm and its analysis remain interesting.
>
> While the greedy heuristic does obtain high objective value, there is a long line of work on the multiobjective problem which aims to improve upon greedy heuristics (Leskovec et al. ‘10; Chekuri et al. ‘10; Orlin et al. ‘18; Udwani ‘18; Tsang et al. ‘19). We contribute to this line of work by providing the first practical algorithm with the best possible approximation guarantee. Although greedy might perform similarly in practice it does not have any theoretical guarantees, as the reviewer points out. Such guarantees are valuable as they ensure that we consistently perform well on any given instance. In particular, we outperform or match greedy in objective value.

---

### Official Review · Reviewer_1Lvm · 2025-03-21

**Overall Recommendation:** 4

**Summary:**

A combinatorial algorithm for multi-objective submodular optimization is developed, that achieves ratio 1-1/e with constant problem under assumptions on the budget and number of colors. This improves over the best ratio achieved by a combinatorial algorithm in prior work. The algorithm requires several novel ideas. A new application of fair centrality is introduced, and an empirical evaluation of the algorithm is included.

**Claims And Evidence:**

Yes.

**Essential References Not Discussed:**

This work isn't essential, but I would appreciate a discussion of [1], as elaborated in strengths and weaknesses.

[1]. Buchbinder, Niv, et al. "Submodular maximization with cardinality constraints." Proceedings of the twenty-fifth annual ACM-SIAM symposium on Discrete algorithms. Society for Industrial and Applied Mathematics, 2014.

**Experimental Designs Or Analyses:**

No.

**Methods And Evaluation Criteria:**

Yes.

**Other Comments Or Suggestions:**

line 107: seting -> setting
It might be a good idea if B > k, or the approximability of the problem were discussed earlier. When reading Problem 3.1, I was wondering about this. If B < k, clearly inapproximable.

**Other Strengths And Weaknesses:**

Strengths:
+ Paper is well written and easy to follow. Contributions are clearly explained and the relationship to prior work is well documented.
+ The main algorithm reminds me a lot of the RandomGreedy algorithm from [1], with some novel differences. RandomGreedy samples a random element from the top k marginal gains to get one with good enough average contribution. This algorithm requires solving an LP to obtain a probability distribution to sample from. Additional ideas are needed to get a bound that holds with constant probability. I do think it would be a good idea to cite [1] and explain the relationship between the algorithms.
+ I found the fair centrality application interesting, and the experimental results clearly show the advantages of the proposed algorithm.
+ I also appreciated an interpretation of fairness in terms of multi-objective optimization, rather than enforcing it through various contraints.

[1]. Buchbinder, Niv, et al. "Submodular maximization with cardinality constraints." Proceedings of the twenty-fifth annual ACM-SIAM symposium on Discrete algorithms. Society for Industrial and Applied Mathematics, 2014.

Weaknesses:
+ Would like to see [1] cited and a discussion of the relationship between the combinatorial algorithm (RandomGreedy) in that paper and this one. Could ideas from RandomGreedy (which works for non-monotone objective) extend to a non-monotone variant of the algorithm in this paper?/

**Questions For Authors:**

- See weaknesses.
- Is there a direction transformation from the notion of fairness as a constraint (as in some of the references discussed in the introduction) to the multi-objective notion of fairness? If so, how do the results in those papers compare with this one?

**Relation To Broader Scientific Literature:**

Improves the state-of-the-art ratio for a combinatorial algorithm for this problem.

**Theoretical Claims:**

I checked the results in the main text through Section 4.1. I even tried to come up with a simpler version of the algorithm that avoided an LP, which didn't end up working.

---

> ### Author Rebuttal · Authors · 2025-03-30
>
> Thank you for your evaluation and the encouraging comments on the strengths of our paper.
>
> > This work isn't essential, but I would appreciate a discussion of [1], as elaborated in strengths and weaknesses.
> > [1]. Buchbinder, Niv, et al. "Submodular maximization with cardinality constraints." Proceedings of the twenty-fifth annual ACM-SIAM symposium on Discrete algorithms. Society for Industrial and Applied Mathematics, 2014.
> > I do think it would be a good idea to cite [1] and explain the relationship between the algorithms.
> > Would like to see [1] cited and a discussion of the relationship between the combinatorial algorithm (RandomGreedy) in that paper and this one. Could ideas from RandomGreedy (which works for non-monotone objective) extend to a non-monotone variant of the algorithm in this paper?
>
> A similarity is that we sample randomly, but our sampling probabilities are carefully chosen to solve the multiobjective problem. One way to see the difference between the algorithms (in terms of the goal and design of either algorithm) is if we run our algorithm for only a single color. Then our algorithm reduces to greedy, showing that our sampling is designed only to enforce good progress across all colors. Their sampling is (intuitively) intended to avoid adding “bad elements” to the solution. However, we believe it could be a promising idea to combine both sampling strategies to obtain an algorithm for non-monotone multiobjective maximization. We will add such a discussion to our paper.
>
> > line 107: seting -> setting It might be a good idea if B > k, or the approximability of the problem were discussed earlier. When reading Problem 3.1, I was wondering about this. If B < k, clearly inapproximable.
>
> In a revised version, we will fix the typo and state after Problem 3.1 that we are interested in the case where the budget is large compared to k ($B>k$).
>
> > Is there a direction transformation from the notion of fairness as a constraint (as in some of the references discussed in the introduction) to the multi-objective notion of fairness? If so, how do the results in those papers compare with this one?
>
> Fairness as a constraint is enforced on the solution set itself. For instance, one can require that the intersection of the solution set with each group (groups are subsets of the ground set) has a certain size. The multiobjective problem is different in that it asks for fairness in the objectives, which corresponds to the notion of max-min fairness on the outcomes. As such, we do not think that one problem can be cast as the other.

---

### Decision · Program_Chairs · 2025-05-01

**Decision:**

Accept (poster)

**Comment:**

The reviewers found the problem studied important, the algorithms elegant and convincing, and the writing clear.